# TRICKS OR TRAPS?
# A DEEP DIVE INTO RL FOR LLM REASONING

**Zihe Liu**[*1,2], **Jiashun Liu**[*1,3], **Yancheng He**[*1], **Weixun Wang**[*1†], **Jiaheng Liu**[4],
**Ling Pan**[3], **Xinyu Hu**[5], **Shaopan Xiong**[1], **Ju Huang**[1], **Jian Hu**[6], **Shengyi Huang**[7],
**Siran Yang**[1], **Jiamang Wang**[1], **Wenbo Su**[1], **Bo Zheng**[1]
[1]Alibaba Group    [2]Beijing Jiaotong University
[3]Hong Kong University of Science and Technology    [4]Nanjing University
[5]Peking University    [6]OpenRLHF    [7]CleanRL

## ABSTRACT

Reinforcement learning (RL) for LLM reasoning has emerged as a prominent research area, marked by a surge in algorithmic innovations and applications. However, critical challenges remain, including the lack of standardized guidelines and fragmented understanding of underlying mechanisms. In addition, inconsistent experimental settings, training data, and model initializations often lead to conflicting conclusions, obscuring key characteristics and creating confusion among practitioners. This paper systematically reviews widely adopted RL techniques through rigorous reproductions and isolated evaluations within a unified open-source framework. We analyze the internal mechanisms, applicable scenarios, and core principles of each technique through fine-grained experiments, including datasets of varying difficulty, model sizes, and architectures. Based on these insights, we present clear guidelines for selecting RL techniques tailored to specific setups and provide a reliable roadmap for practitioners navigating the RL for the LLM domain. Finally, we show that a minimalist combination of two techniques can unlock the learning capability of critic-free policies with a vanilla PPO loss. The results demonstrate that our simple combination consistently improves performance, surpassing strategies such as GRPO and DAPO.

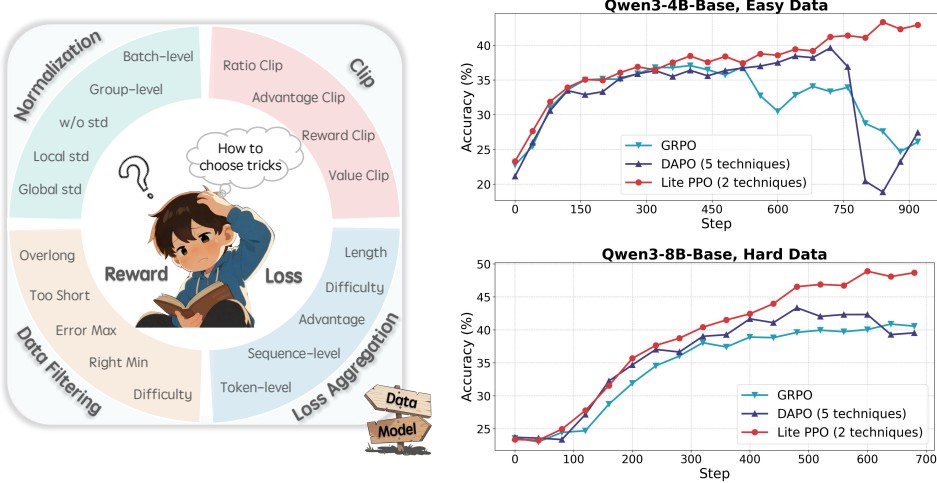

Figure 1: **Left**: The proliferation of RL optimization techniques, coupled with diverse initialized models and data, has raised barriers to practical adoption. **Right**: We establish detailed application guidelines via dissecting internal mechanisms of widely-used tricks, and introduce Lite PPO, a minimalist two-technique combination that enhances learning capacity in critic-free policies with vanilla PPO loss. The average accuracy is calculated across six mathematical benchmarks.

---

* Equal Contribution.    † Corresponding to: Weixun Wang <weixun.wwx@taobao.com>.

# 1 INTRODUCTION

Recent breakthroughs in large language models (LLMs) such as OpenAI o1 (Wu et al., 2024) and DeepSeek R1 (Shao et al., 2024) have positioned reinforcement learning (RL) as a key driver in unlocking advanced reasoning capabilities in LLMs. This is particularly evident in challenging reasoning tasks like mathematical problem solving (He et al., 2025b) and code generation (Zhuo et al., 2025), where RL has demonstrated the potential to elevate LLM performance beyond what pre-training alone can achieve. Such an emerging trend has sparked widespread interest within the research community in the direction of "RL for LLM" (or RL4LLM).

However, this rapid progress is shadowed by a lack of usage guidelines (Huang et al., 2024b) and in-depth understanding of underlying mechanisms. For instance, GRPO (Shao et al., 2024) advocates group-level normalization for stability, while REINFORCE++ (Hu et al., 2025) favors batch-level. Moreover, GRPO incorporates variance in normalization, yet Dr. GRPO (Liu et al., 2025b) explicitly recommends removing variance normalization to prevent bias. Such contradictory and chaotic phenomena underscore the fragmented understanding and inconsistent recommendations within the RL4LLM community. A likely cause for the above phenomenon is that the experimental settings, training data, and initialization of the existing work all have significant differences, which may also cause deviations in the summary of the conclusions.

Apart from the confusion caused by the intrinsic differences of similar techniques, the numerous and seemingly orthogonal techniques, including *Normalization, Clip, and Overlong Filtering*, also increase the complexity of algorithm application in practice. Practitioners face non-trivial challenges in identifying an effective combination from a wide range of techniques to unlock the learning capacity of LLMs in specific scenarios. These ambiguities have naturally triggered a key requirement of practitioners:

What scenarios are the existing techniques respectively suitable for? Is there a simple and generalizable combination that can be used to enhance policy optimization?

Following established RL analysis practices (Andrychowicz et al., 2020; Engstrom et al., 2020; Huang et al., 2024b), we systematically evaluate popular RL techniques through reproducible experiments on a unified open-source framework. Our comprehensive setup spans datasets of varying difficulty, different model sizes, and multiple model types, supported by **over 160 independent RL training experiments** to ensure robust and statistically meaningful conclusions. We also delve into the theoretical foundations, implementation, and recommended applications of each technique, as summarized in Figure 1. Our findings show that most RL methods are highly sensitive to factors like model type, data distribution, reward design, and hyperparameters. Notably, we demonstrate that combining just two techniques—*advantage normalization (group mean, batch std)* and *token-level loss aggregation*—is sufficient to maximize the potential of critic-free policies with vanilla PPO loss, outperforming mainstream RL4LLM approaches that rely on extra components. Our key contributions are:

1. Removing the standard deviation when reward distributions are highly concentrated enhances the stability and effectiveness of model training. (§4.1.2)

2. Group-level mean and batch-level standard deviation enable further robust normalization. (§4.1.3)

3. Clip Higher promotes high-quality exploration for aligned models. (§4.2.1)

4. There appears to be a "scaling law" between the performance and the upper bound of the clipping on the small-sized model. (§4.2.3)

5. Compared to sequence-level loss aggregation, token-level aggregation is effective on base models but shows limited improvement on aligned models. (§4.3.1)

6. Overlong filtering enhances accuracy and clarity for short-to-medium reasoning tasks but provides limited benefits for long-tail reasoning. (§4.4.1)

7. Two techniques may unlock learning capacity in critic-free policies based on vanilla PPO loss. (§5)

## 2 Preliminaries

A variety of practical techniques have been introduced to stabilize optimization, reduce variance, and accelerate the convergence of LLMs on reasoning tasks. We categorize commonly used techniques as follows: **(1) Baseline Design.** Baselines are crucial to reduce variance in the estimation of policy gradients, with recent advances including using the group-mean reward (Shao et al., 2024) or computing the baseline for each sample as the average gradient estimate from other samples in the group (Ahmadian et al., 2024; Kool et al., 2019). **(2) Clipping Strategies.** Clipping controls excessive policy updates by constraining rewards, advantages, or ratios, and the *Clip Ratio Higher* method enhances exploration by relaxing the upper bound in PPO's ratio clipping (Yu et al., 2025). **(3) Normalization Strategies.** Normalization of rewards or advantages stabilizes gradient magnitudes, with representative methods including *Batch-level* (Hu et al., 2025), *Group-level* (Shao et al., 2024; Ahmadian et al., 2024), and *Reward Shift without Standard Deviation* (Liu et al., 2025b). **(4) Filtering Strategies.** Filtering excludes uninformative or undesirable samples before gradient computation, including *Overlong Filtering* for excessive lengths (Yu et al., 2025), *Error Max Clip Mask* and *Right Min Clip Mask* for extreme correctness or errors, and *Difficulty Mask* to exclude samples outside a target difficulty range (Yu et al., 2025; Zhang et al., 2025; Chu et al., 2025). **(5) Loss Aggregation Granularity.** The formulation of loss aggregation determines each token's contribution to the overall objective, with common approaches including *Sequence-level Loss* and *Token-level Loss*, the latter computes per-token advantages to mitigate length bias. **(6) Additional Loss Functions.** Auxiliary losses can complement the primary objective and regularize training. *KL Loss* (Yu et al., 2025; Liu et al., 2025b) constrains divergence from a reference policy, while *SFT Loss* (Zhang & Zuo, 2025) incorporates supervised fine-tuning objectives to preserve alignment. **(7) Reward Design.** Shaping the reward function can guide desired output properties, with common strategies including *Length Penalty* to discourage overly long outputs, *Formatting Reward* to promote structured outputs, and *Length-Dependent Accuracy Reward* that combines correctness with output length.

## 3 Experimental Designs

### 3.1 Experimental Setup

**Training Algorithm:** We utilize the open-sourced ROLL framework (Wang et al., 2025), an efficient and scalable platform specifically designed for reinforcement learning optimization in LLMs, to conduct all experiments. In addition, we adopt the PPO loss (Schulman et al., 2017), with advantage values computed using the REINFORCE algorithm (Sutton et al., 1999) as the unified RL baseline. To ensure consistency with prior research, we set the global batch size to $1024$ by using a rollout batch size of $128$ and sampling $8$ responses per prompt, with a maximum response length of $8192$ tokens. The learning rate is set to $1e-6$. For text generation, we use a top_p value of $0.99$, a top_k value of $100$, and a temperature of $0.99$.

**Base Models:** To comprehensively evaluate reinforcement learning (RL) techniques across parameter scales, our experiments cover two model sizes: Qwen3-4B and Qwen3-8B. For each model size, we include both non-aligned pre-trained versions (Qwen3-4B-Base and Qwen3-8B-Base) and aligned versions, enabling assessment RL gains from different initialization conditions.

**Training Datasets:** To ensure reproducibility and fairness, we exclusively use open-source datasets for training, including *SimpleRL-Zoo-Data* (Zeng et al., 2025) and *Deepmath* (He et al., 2025b). To comprehensively examine how problem difficulty (e.g., Easy, Medium, and Hard; detailed in Appendix B.3) affects the performance of the RL technique, we randomly sample $5,000$ entries from the datasets. Figure 2 visualizes the difficulty across the training dataset assessed by GPT-4o (Hurst et al., 2024).

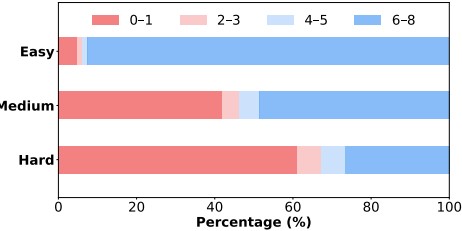

Figure 2: Number of correct responses under 8 rollout iterations across datasets.

---

Open source RL framework: `https://github.com/alibaba/ROLL`

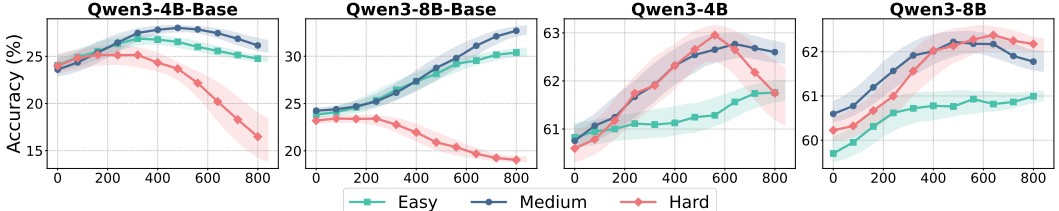

Figure 3: Test accuracy over training iterations is compared for different models trained on different datasets. The shaded regions represent the oscillation amplitude as mean $\pm$ (std_multiplier $\times$ std), with curves smoothed using an 11-step moving window and exponential smoothing ($\alpha = 0.8$).

**Evaluation Benchmark:** All the experiments are conducted on six math datasets: MATH-500 (Hendrycks et al., 2021), OlympiadBench (He et al., 2024), MinervaMath (Lewkowycz et al., 2022), and subsets of standardized examinations (AIME24-25, AMC23). These datasets span a broad complexity spectrum from basic arithmetic to competition-level mathematics, enabling a comprehensive evaluation of reasoning capabilities.

## 3.2 BASELINE RESULTS

**Impact of Data Difficulty on Training Dynamics** We investigate how data difficulty influences the training dynamics of Qwen3 models. Specifically, we analyze the training convergence patterns through accuracy trajectories and generalization gaps, across three tiers of complexity (*Easy, Medium, Hard*). The detailed learning curves in Figure 3 show that the model exhibits markedly different accuracy trajectories across training sets of different difficulty levels. When focusing on the differences in learning efficiency between the unaligned Base model and the aligned model under the same experimental setting, the aligned models exhibited substantially higher initial accuracy, but additional learning yielded only modest gains, with accuracy improving by roughly 2%. Additional details regarding the baseline's training accuracy and response length can be found in Appendix C.

## 4 ANALYSIS

### 4.1 NORMALIZATION

Advantage normalization is a standard technique for stabilizing RL training in language models by reducing gradient variance (Zheng et al., 2023), yet implementations vary. GRPO (Shao et al., 2024) and RLOO (Ahmadian et al., 2024; Kool et al., 2019) use group-level normalization to promote intra-context competition, while REINFORCE++ (Hu et al., 2025) adopts batch-level normalization to mitigate overfitting and reward hacking in low-diversity settings. Formally, given a prompt $x$ with $K$ sampled responses and corresponding rewards $\{r_k\}_{k=1}^K$, and denoting $r_i$ as the reward of the $i$-th response in a rollout batch of $N$ prompts with $K$ responses each, the group-level and batch-level normalized advantages are:

$$A_k^{\text{group}} = \frac{r_k - \text{mean}(\{r_j\}_{j=1}^K)}{\text{std}(\{r_j\}_{j=1}^K)} \quad A_i^{\text{batch}} = \frac{r_i - \text{mean}(\{r_j\}_{j=1}^{N*K})}{\text{std}(\{r_j\}_{j=1}^{N*K})} \quad (1)$$

#### 4.1.1 ADVANTAGE NORMALIZATION IS SENSITIVE TO REWARD MECHANISMS

To systematically evaluate the impact of advantage normalization on PPO variants with a value function using the Monte Carlo return target, we conducted experiments under a unified training framework, exploring three settings: **no normalization**, **batch-level normalization**, and **group-level normalization**. When analyzing the performance in Figure 4, it can be concluded that both advantage normalization techniques can significantly influence the model's convergence speed, performance stability, and final outcomes.

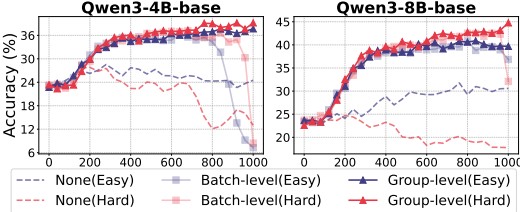

Figure 4: Accuracy comparison of Base models with different normalization techniques cross Easy and Hard datasets.

Specifically, on both model sizes, group-level normalization consistently achieves more stable training dynamics and higher final performance compared to both batch-level normalization and no normalization. Batch-level normalization exhibits high sensitivity to reward distribution skew, often leading to performance collapse under an imbalanced batch situation, where a few outlier samples dominate the advantage estimates.

### 4.1.2 IMPACT OF THE STANDARD DEVIATION TERM IN ADVANTAGE NORMALIZATION

> **Takeaway 1**
>
> **Removing the standard deviation** when reward distributions are highly concentrated (e.g., easy training dataset) enhances the stability and effectiveness of model training.

We found when model responses within a prompt group yield highly similar rewards, e.g., when the responses are almost all correct or all incorrect, the resulting standard deviation becomes extremely small. In such cases, dividing by this small standard deviation during normalization can excessively amplify gradient updates, causing the model to overemphasize tasks of extreme difficulty, a phenomenon similar to "difficulty bias" (Liu et al., 2025b).

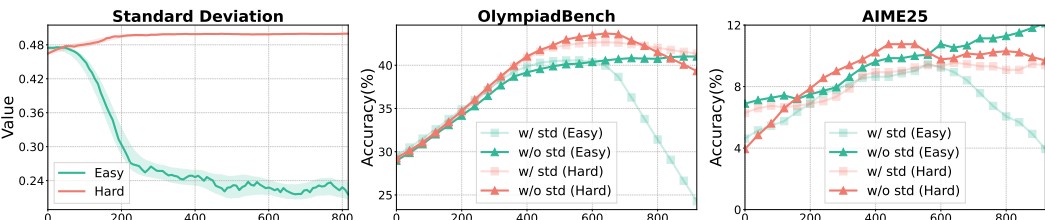

Figure 5: **Left:** Standard deviation variations during training on easy and hard datasets. **Middle and Right:** Test accuracy with and without batch-level standard deviation across easy and hard dataset.

To test whether the standard deviation term is the critical factor driving differences in normalization performance, we employ the batch-level calculation, which exhibited unstable performance in the previous section, to calculate the mean of advantage, and conduct ablation experiments on the standard deviation term. This can be formalized as:

$$A_k^{\text{std}^{\neg}} = r_k - \text{mean}(\{r_j\}_{j=1}^{K}). \tag{2}$$

Our experiments reveal that, when training on easy data, the policy quickly converges to consistent behaviors, resulting in a highly concentrated reward distribution and a rapid decline in standard deviation. In this scenario, using standard deviation-based advantage normalization can cause the denominator to become extremely small, excessively amplifying reward and advantage values. This leads to abnormally large gradients, destabilizes training, and may even cause gradient explosions. These findings empirically demonstrate that the standard deviation term plays a crucial role in advantage normalization.

In summary, our experiments and analysis underscore that, in scenarios where reward distributions are highly concentrated, omitting the standard deviation from advantage normalization effectively prevents abnormal gradient amplification, thereby improving the stability and robustness of model training. However, for tasks characterized by inherently higher reward variance, either normalization approach is generally sufficient to maintain stable optimization.

### 4.1.3 RECONSTRUCT A ROBUST NORMALIZATION TECHNIQUE

> **Takeaway 2**
>
> Calculating the mean at the local (group) level and the standard deviation at the global (batch) level enables more robust reward shaping.

Section 4.1.2 highlights the critical role of standard deviation in advantage normalization. This raises the question: is there a more robust and effective combination of mean and standard deviation for reward shaping? To explore this, we adopted the stable group-level mean calculation method demonstrated in section 4.1.1, paired with two approaches for computing the standard deviation: local (group-level) and global (batch-level). We then evaluated the performance of these combinations across two model sizes.

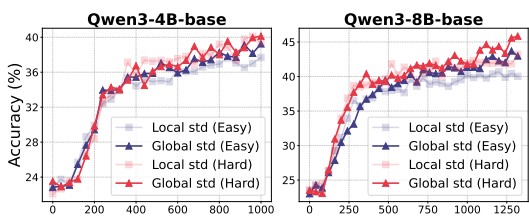

Figure 6: Accuracy comparison of base models with different standard deviation calculation.

The results, presented in Figures 6, reveal that global-level calculation exhibits a clear advantage. We attribute this to the batch-level standard deviation providing stronger normalization by effectively reducing gradient magnitudes, thereby preventing excessive policy updates. This approach aligns more effectively with the biased reward signals common in sparse rewards and coarse-grained advantage fitting, resulting in more stable and robust learning behavior. Furthermore, our experimental results support a claim from Hu et al. (2025) that batch-level normalization, or even subtracting the local mean and dividing by the batch standard deviation in certain scenarios, performs better.

## 4.2 CLIP-HIGHER

The Clip mechanism, while improving PPO training stability (Huang et al., 2024a), can cause issues in LLM-based text generation by overly suppressing low-probability tokens (Yu et al., 2025), leading to entropy collapse—where model output becomes overly deterministic and loses diversity (Jin et al., 2024). This reduction in entropy reinforces high-probability patterns and shrinks exploration, impairing performance on tasks requiring novel reasoning.

$$J_{DAPO}(\theta) = (r_{i,t}(\theta), 1 - \varepsilon_{low}, 1 + \varepsilon_{high}). \tag{3}$$

Here, a greater upper bound $\varepsilon_{high}$ enables more exploration for low-probability tokens, mitigating entropy collapse. However, there's limited analysis on when and how to set this upper bound effectively. In this section, we address these gaps with targeted experiments and recommendations.

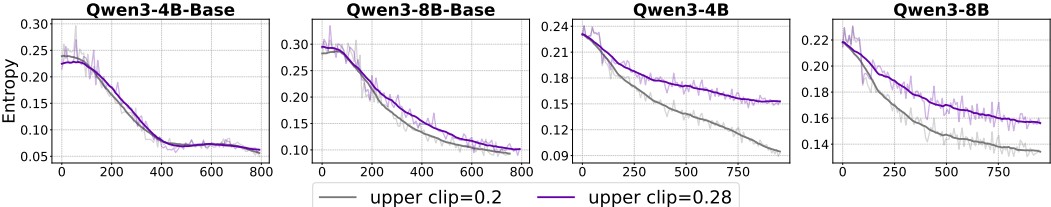

Figure 7: Entropy comparison across different models with Clip-Higher. **A higher clip upper bound can mitigate the entropy drop in aligned models**.

### 4.2.1 IN WHICH SETTINGS SHOULD WE CLIP HIGHER

> **Takeaway 3**
>
> For models with stronger fundamental reasoning abilities, increasing the clip higher parameter is more likely to facilitate exploration of better solution paths.

As illustrated in Figure 7, experimental results indicate that the impact of increasing the upper clipping bound $\varepsilon_{high}$ is model-dependent. For the base models, adjusting the upper clipping value yields minor effects on policy entropy. In contrast, aligned models exhibit a markedly different response: raising the upper clipping bound notably slows the entropy collapse, leading to consistent performance improvements in downstream evaluation metrics.

This disparity is mainly due to the base models' low clipping rate (around 0.003), which results in minimal policy updates and limited exploration because of their simple policy expressiveness. As a

result, increasing the clipping upper bound has little effect on learning outcomes. In contrast, aligned models demonstrate improved reasoning and generalization abilities due to advanced post-training techniques (Yang et al., 2025). Figure 18 in Appendix D, illustrates that, compared to base models, aligned models initially have fewer high-probability tokens. Therefore, a higher clipping upper bound reduces token probability disparity and mitigates entropy collapse. It expands the range of permissible policy updates, promoting more diverse action sampling and enhanced exploration during training. This mechanism maintains higher entropy while increasing the likelihood of finding optimal solutions, as shown by improved evaluation metrics.

### 4.2.2 ANALYZING THE EFFECTIVENESS OF CLIP-HIGHER FROM A LINGUISTIC PERSPECTIVE

> **Takeaway 4**
>
> **Traditional clipping** may constrain innovative reasoning structure generation. **Clipping higher** allows the model to explore a broader range of discourse reasoning structures.

Building on our token-level demonstration of Clip-Higher's behavior in section 4.2.1, we now analyze its impact on reasoning logic through token-level linguistics. As shown in Figure 20 in Appendix D, an upper bound of 0.2 imposes strict constraints on policy updates, primarily affecting connective tokens like *therefore*", *if*", and "*but*" by limiting large probability deviations. This leads to fewer opportunities for the model to generate innovative or diverse logical structures. By increasing the upper bound to 0.28, the model gains more flexibility, resulting in fewer tokens being clipped and a shift in clipping focus toward high-frequency function words such as "*is*", "*the*", and ",". This encourages richer reasoning paths while ensuring sentence stability through selective clipping of common function words.

### 4.2.3 HOW TO SET THE UPPER BOUND FOR ADVANTAGE CLIPPING

> **Takeaway 5**
>
> There appears to be a "scaling law" between the performance and the upper bound of the clipping on the **small-sized model**, which does not exist on **larger models**.

Section 4.2.1 shows that Clip-Higher improves aligned models. While most works use the default clip upper bound of 0.28 (Yu et al., 2025), we believe that different models have different preferences for this parameter. To verify this conjecture, we uniformly evaluate different clip upper bounds ranging from 0.2 to 0.32 on two model sizes. As shown in Figure 8, the 4B model improves progressively with higher bounds, peaking at 0.32. In contrast, the 8B model performs best at 0.28, with no consistent gain beyond that.

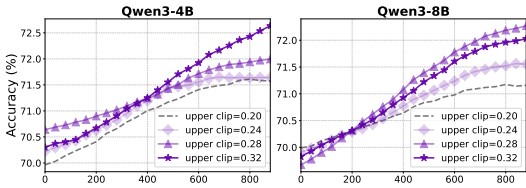

Figure 8: Test accuracy of aligned models (trained on medium data) with various clipping upper bounds.

### 4.3 LOSS AGGREGATION

The strategy of loss aggregation directly determines the contribution of each sample or token to the overall gradient during optimization (Liu et al., 2025a). Common strategies include token-level and sequence-level aggregation. The sequence-level aggregation adopted by GRPO (Shao et al., 2024) first averages the loss across all tokens within each sample, then averages these per-response losses across the batch, thereby assigning equal weight to each response regardless of its length. However, Yu et al. (2025) highlights a flaw in this method: longer responses possess a diminished influence per token on the total loss, hindering the model's ability to learn effectively from longer, complex responses. This can reduce the model's capacity to learn from long, complex answers, and may bias optimization toward brevity, since shorter correct responses receive larger gradient updates, while longer incorrect responses are insufficiently penalized (Liu et al., 2025b).

### 4.3.1 DOES TOKEN-LEVEL LOSS AGGREGATION SUIT ALL SETTINGS?

> **Takeaway 6**
>
> Compared to sequence-level calculation, token-level loss proves to be more effective on Base models, while showing limited improvement on Instruct models.

To systematically evaluate loss aggregation strategies, we compare sequence-level and token-level loss aggregation on both base and aligned versions of Qwen3-8B. As illustrated in Figure 9, token-level loss improves convergence and peak accuracy for base models by balancing token contributions, particularly on challenging data. In contrast, this advantage disappears in aligned models, where sequence-level aggregation achieves better convergence speed and final performance. Analysis suggests that aligned models already exhibit stable reasoning, making fine-grained token weighting unnecessary or harmful. In these cases, sequence-level aggregation better preserves the structure and consistency of high-quality, aligned outputs. These findings highlight that optimal aggregation is model-dependent: token-level suits base models; response-level is preferable for instruction-tuned ones.

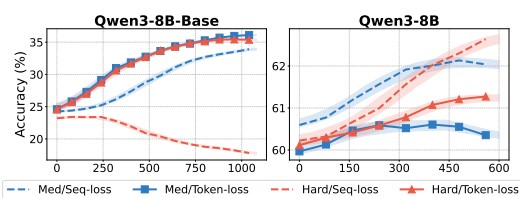

Figure 9: Accuracy comparison between sequence-level loss and token-level loss. Results are reported on both Medium and Hard Datasets.

## 4.4 OVERLONG FILTERING

To improve efficiency, LLMs often use fixed maximum generation lengths for truncation (Chen et al., 2025; Team et al., 2025). However, this may truncate multi-step reasoning early in training, causing incomplete outputs to be mislabeled as negatives and introducing harmful noise. Overlong filtering (Yu et al., 2025) mitigates this by masking rewards of over-length responses, preserving robustness and reasoning quality (He et al., 2025a). Yet, its sensitivity to mask thresholds remains underexplored, leaving optimal settings unclear.

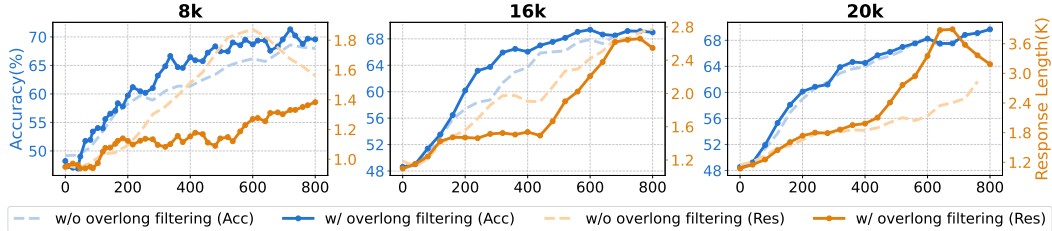

Figure 10: Total test accuracy and response length of Qwen3-8B-Base over training iterations under different maximum generation lengths.

### 4.4.1 WHEN TO USE THE OVERLONG FILTERING

> **Takeaway 7**
>
> Overlong filtering shows limited effectiveness on long-tail reasoning tasks; however, it can enhance the accuracy and clarity of responses in medium and short-length reasoning tasks.

The results in Figure 10 shows that overlong filtering improves learning at a short threshold ($8k$), but benefits diminish at $20k$. Response length analysis reveals the cause: under $20k$, filtered models generate longer outputs than the vanilla policy; at $8k$, responses become shorter. As Figure 11 (Left) shows, in the $20k$ setting, frequent clipping occurs due to repetition or non-termination—signs of degenerate generation—indicating the mask removes unproductive outputs. In contrast, the $8k$ threshold also filters extended but valid reasoning, encouraging conciseness and reducing verbosity.

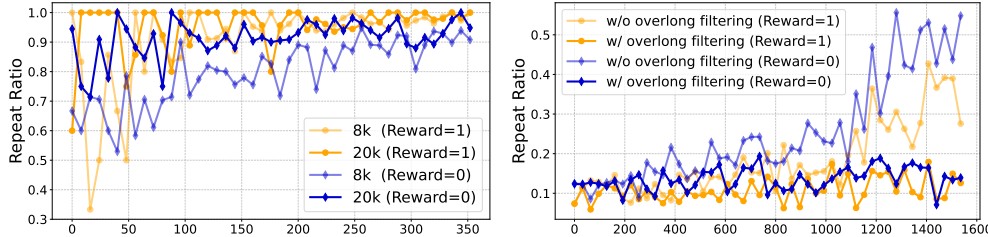

Figure 11: **Left:** Repeat ratios for correct (reward = 1) and incorrect (reward = 0) generations under varying maximum generation lengths. **Right:** Repeat ratios in truncated samples with vs. without overlong filtering. Repetition rate statistics are provided in Appendix E.1.

As illustrated in Figure 11 (Right), during RL training, the proportion of "repetitive but unable to terminate" overlong samples increases, indicating degraded EOS modeling and leading to output redundancy and termination failures. With the overlong mask, this proportion drops and remains low, enabling the model to better distinguish "generation completed" from "truncated" samples, avoiding invalid learning. More importantly, the mechanism helps policies accurately model termination, preventing them from mistakenly penalizing unfinished outputs as negative examples.

## 5   A SIMPLE COMBINATION: LITE PPO

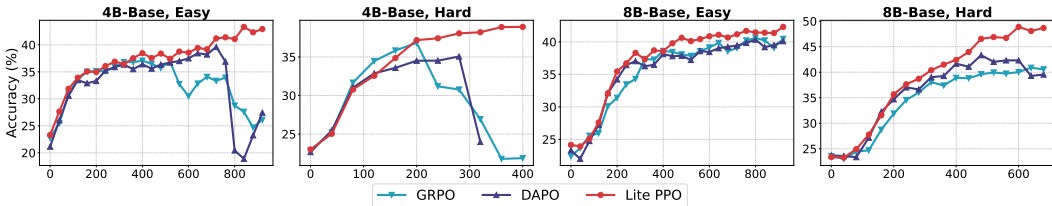

Figure 12: Test accuracy of non-aligned models trained via three RL methods, i.e., Lite PPO (ours), GRPO (Shao et al., 2024) and DAPO (Yu et al., 2025).

Building on the in-depth mechanism analysis, we derive two practical guidelines for non-aligned models: (i) For small and medium-sized base models, advantage normalization (Section 4.1.3) significantly boosts performance by transforming sparse rewards into robust signals via group-mean and batch-std normalization. (ii) Token-level loss aggregation is also highly effective, as shown in Section 4.3.1, particularly for base model architectures.

Therefore, we integrate both techniques, called Lite PPO, into non-aligned models that use the vanilla PPO loss without the critic. As shown in Figure 17, Lite PPO outperforms GRPO and DAPO, a technique-heavy method with *Group-level Normalization, Clip-Higher, Overlong Reward Shaping, Token-level Loss, Dynamic Sampling*. Specifically, Lite PPO exhibits a stable upward trend on non-aligned models, while other policies collapse after peaking. This advantage arises from the normalization in Takeaway 2, which mitigates interference from homogeneous reward distributions in mixed datasets. Additionally, this gain stems from adopting token-level loss aggregation, which is more effective for base models.

## 6   CONCLUSION

We present a systematic evaluation of RL techniques for LLMs under a unified framework, addressing fragmentation in methodology and practice. By analyzing normalization, clipping, and filtering, we reveal that simplicity outperforms complexity: Lite PPO, combining only two core techniques, surpasses heavily engineered algorithms. Our findings highlight the importance of context-aware design and challenge the trend of over-complication in RL4LLM. We provide actionable guidelines for technique selection and advocate for standardized, reproducible practices that balance theoretical soundness with practical efficiency.

## 7 ETHICS STATEMENT

This work adheres to the ICLR Code of Ethics. In this study, no human subjects or animal experimentation was involved. All datasets used, including Easy, Medium and Hard datasets, were sourced in compliance with relevant usage guidelines, ensuring no violation of privacy. We have taken care to avoid any biases or discriminatory outcomes in our research process. No personally identifiable information was used, and no experiments were conducted that could raise privacy or security concerns. We are committed to maintaining transparency and integrity throughout the research process.

## 8 REPRODUCIBILITY STATEMENT

We have made extensive efforts to ensure the reproducibility of our work. The baseline models we evaluated are detailed in Section 3.2 and Appendix D. We provide not only the specific models and API versions, but also the exact sampling parameters we used. The evaluation metrics are described in Section 3.1. We will release our benchmark dataset and evaluation code upon paper acceptance to facilitate reproduction and future research by the community.

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

## A    LLM Usage Statement

Large Language Models (LLMs) were used to aid in the writing and polishing of the manuscript. Specifically, we used an LLM to assist in refining the language, improving readability, and ensuring clarity in various sections of the paper. The model helped with tasks such as sentence rephrasing, grammar checking, and enhancing the overall flow of the text.

It is important to note that the LLM was not involved in the ideation, research methodology, or experimental design. All research concepts, ideas, and analyses were developed and conducted by the authors. The contributions of the LLM were solely focused on improving the linguistic quality of the paper, with no involvement in the scientific content or data analysis.

The authors take full responsibility for the content of the manuscript, including any text generated or polished by the LLM. We have ensured that the LLM-generated text adheres to ethical guidelines and does not contribute to plagiarism or scientific misconduct.

## B    Detailed Experimental Setup

### B.1    Parameters

We employ ROLL, a user-friendly and efficient open-source reinforcement learning framework, to implement our pipeline. Subsequently, the key parameters observed during the training process are presented as follows. See our code config file for more details on the parameters.

### B.2    Prompt

In this work, we incorporate the following instruction into the system prompt to encourage the model to better demonstrate its reasoning process: **"Please reason step by step, and put your final answer within \boxed{}."** This setting is designed to guide the model to perform step-by-step reasoning and explicitly present the final answer in the form of \boxed{}, thereby enhancing the clarity and readability of the output.

### B.3    Training Datasets

To ensure reproducibility and fairness, we exclusively use open-source datasets for training, including *SimpleRL-Zoo-Data* (Zeng et al., 2025) and *Deepmath* (He et al., 2025b). To comprehensively examine how problem difficulty affects the RL technique's performance, we randomly sample from the datasets, removing an excessive proportion of examples whose ground-truth label is simply "True" or "False". This adjustment addresses the **ostensible positive phenomenon**, where models produce correct binary answers from erroneous reasoning chains, thereby introducing noisy supervision that compromises training quality (please refer to Appendix E.2 for case studies).

- Easy Data : We randomly sample $5,000$ entries from SimpleRL-Zoo-Data-Easy, which consists of problems drawn from GSM8K and MATH-500-level1.

- Medium Data: We select the $5,000$ easiest examples from the *DeepMath-103k* dataset, based on their assigned difficulty annotations.

- Hard Data: We randomly sample $5,000$ entries from *DeepMath-103k*, with sampling probability proportional to each entry's assigned difficulty level.

```
seed: 42
max_steps: 500
save_steps: 20
logging_steps: 1
eval_steps: 1

rollout_batch_size: 128
prompt_length: 1024
response_length: 8000

ppo_epochs: 1
adv_estimator: "reinforce"
init_kl_coef: 0.0
async_generate_level: 1

actor_train:
  training_args:
    learning_rate: 1.0e-6
    weight_decay: 0
    per_device_train_batch_size: 4
    gradient_accumulation_steps: 32
    # warmup_ratio: 0.1
    warmup_steps: 50
    num_train_epochs: 50
  ...

actor_infer:
  generating_args:
    max_new_tokens: ${response_length}
    top_p: 0.99
    top_k: 100
    num_beams: 1
    temperature: 0.99
    num_return_sequences: 8
  ...
```

# C  DETAILED EXPERIMENTAL RESULTS

## C.1  BASELINE

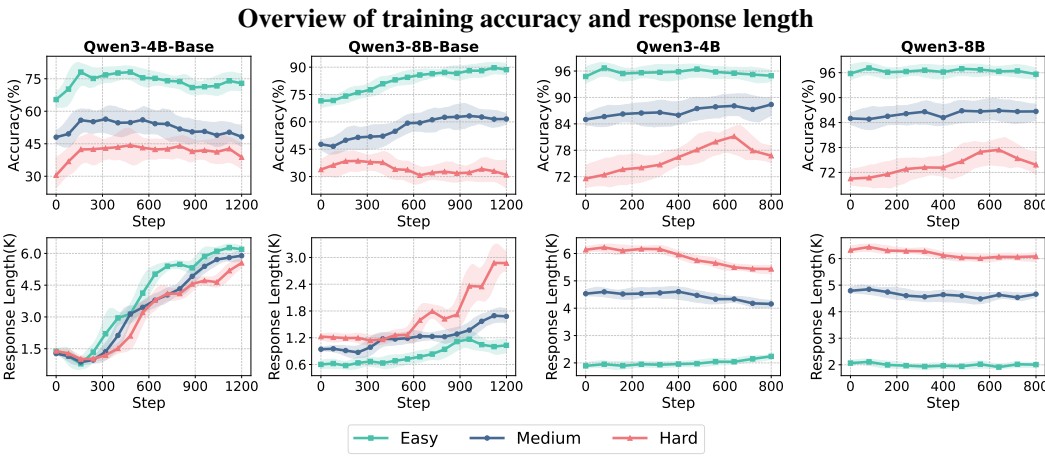

Figure 13: Test accuracy and response length of four model variants: `Qwen3-4B-Base`, `Qwen3-8B-Base`, `Qwen3-4B`, and `Qwen3-8B` across different data difficulty.

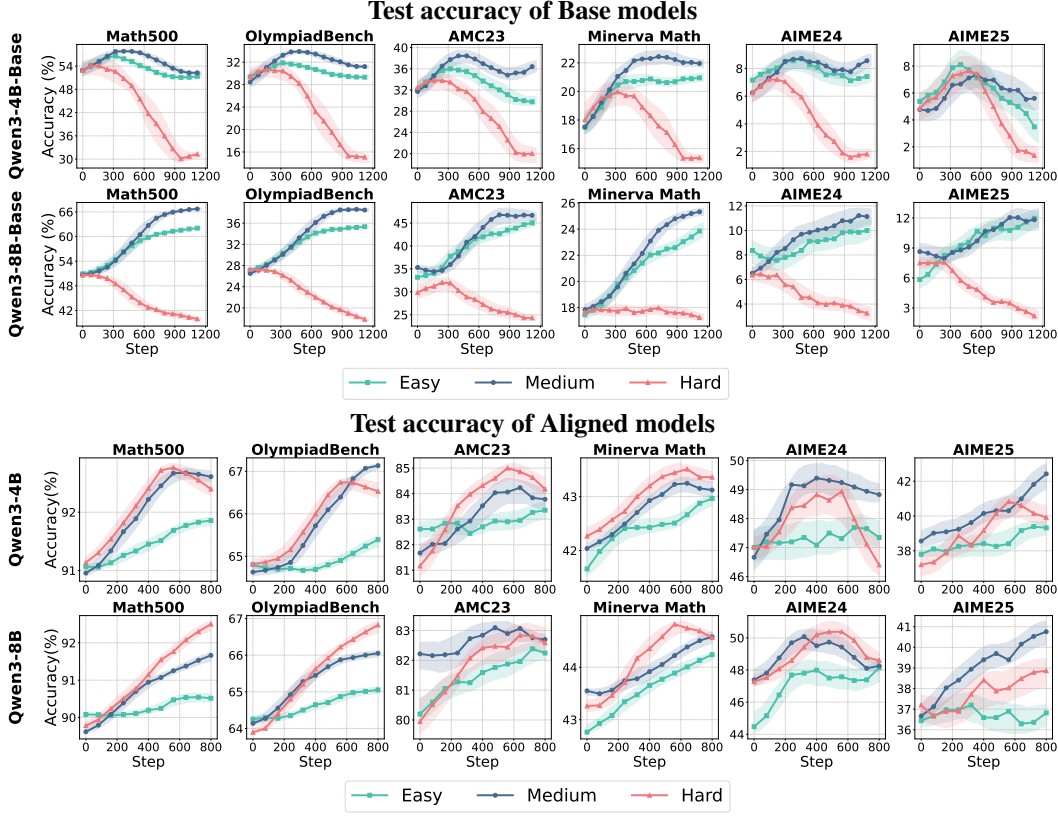

Figure 14: **Middle 2 rows**: Accuracy over training iterations of Base models. The first row presents results of `Qwen3-4B-Base`. The second row shows results of `Qwen3-8B-Base`. **Bottom 2 rows**: Accuracy over training iterations of aligned models. The first row presents results of `Qwen3-4B`, while the second row shows results of `Qwen3-8B`.

## C.2 RESULTS OF ALIGNED MODEL

Following our analysis in section 4.2.1 that Clip Higher provides stronger benefits on aligned models, we instantiated a variant of LitePPO that combines mixture normalization and Clip Higher, trained on both the Easy and Hard datasets.

As shown in Tables 1 and 2, our simple recipe significantly improves the reasoning ability of the aligned model and consistently outperforms DAPO, particularly when trained on the hard dataset. This confirms that LitePPO remains efficient for strong, instruction-tuned models.

|          | MATH-500 | OlympiadBench | MinervaMath | AIME24 | AIME25 | AMC23 | Average |
|----------|----------|---------------|-------------|--------|--------|-------|---------|
| DAPO     | 90.57    | 64.82         | 43.43       | 47.92  | 37.54  | 79.69 | 60.66   |
| Lite PPO | **91.43** | **66.01**    | **44.72**   | **49.17** | **39.81** | **81.88** | **62.17** |

Table 1: Results using **Qwen3-8B (aligned model)** trained on the **Easy dataset**.

|          | MATH-500 | OlympiadBench | MinervaMath | AIME24 | AIME25 | AMC23 | Average |
|----------|----------|---------------|-------------|--------|--------|-------|---------|
| DAPO     | 93.02    | 69.15         | 45.73       | 52.50  | 39.17  | 85.31 | 64.17   |
| Lite PPO | **94.77** | **71.12**    | **46.60**   | 49.58  | **47.50** | **92.19** | **66.96** |

Table 2: Results using **Qwen3-8B (aligned model)** trained on the **Hard dataset**.

## C.3 VALIDATING METHOD EXTENSIBILITY

We have conducted additional experiments on Llama3-8B, we recorded the average score among six benchmarks,i.e., AIME 24, AMC23, GSM8k, MATH-500, Minerva Math, OlypiadBench. The results in Table 3 show that, compared to GRPO and DAPO, our Lite PPO demonstrates the best average performance, indicating the portability of our method..

|          | MATH-500 | OlympiadBench | AIME 24 | AMC23 | GSM8k | Minerva Math | Average |
|----------|----------|---------------|---------|-------|-------|--------------|---------|
| GRPO     | 19.8     | 8.3           | 3.7     | 18.2  | 77.3  | 8.5          | 22.63   |
| DAPO     | 25.1     | 7.6           | 4.6     | 18.4  | 77.1  | 9.2          | 23.67   |
| Lite PPO | 27.3     | 9.7           | 9.1     | 17.9  | 79.6  | 9.6          | 25.53   |

Table 3: Results on Llama3-8B.

We have conducted additional experiments to evaluate the generalization performance of our method. Specifically, we test the Qwen3-8b-Base trained by LitePPO on several out-of-domain reasoning tasks, including coding (LCB-v5, LCB-v6), Interdisciplinary QA (GPQA), and Language Understanding (MMLU-Pro). The results in Table 4 show that Lite PPO achieves the pass@1 score of 25.08 on LCB-v5, 19.71 on LCB-v6, 46.63 on GPQA, and 62.38 on MMLU-Pro, demonstrating our method's excellent generalization ability.

|          | LCB-v5 | LCB-v6 | GPQA  | MMLU-Pro |
|----------|--------|--------|-------|----------|
| GRPO     | 22.94  | 18.28  | 42.49 | 61.84    |
| DAPO     | 24.01  | 19.00  | 45.08 | 60.60    |
| **LitePPO** | **25.08** | **19.71** | **46.63** | **62.38** |

Table 4: Pass@1 score on other reasoning modalities.

## D CASE STUDY OF CLIP HIGHER

We present a comparison of token distributions between the base model and the aligned model.

As shown in Figure 18, compared to the base model, the aligned model has very few preferred tokens with high probability in the initial stage. Therefore, a higher clipping upper bound can effectively bridge the probability gap between tokens and alleviate the entropy collapse. For these models, raising the upper bound expands the permissible range of policy updates, which in turn facilitates more diverse action sampling and enhances exploratory behavior during training. This mechanism preserves higher entropy while simultaneously increasing the probability of identifying optimal solutions, as evidenced by improved evaluation metrics.

Building on our token-level demonstration of Clip-Higher's behavior in section 4.2.1, we now analyze its impact on reasoning logic through token-level linguistics. As illustrated in Figure 20, setting an upper bound to 0.2 imposes stringent constraints on policy updates by limiting substantial probability deviations for individual tokens. Under these stricter conditions, our analysis reveals that clipping predominantly affects connective tokens such as "*therefore*", "*if*", and "*but*". These tokens frequently appear at the beginnings of sentences, serving as key semantic markers or transition words within dialog generation. Such connectors often introduce new directions in reasoning. However, their probability ratios between updated and old policies frequently exceed clipping thresholds, triggering aggressive suppression in PPO optimization. While this traditional clipping ensures stability in the overall token distribution, it may restrict the model's capacity to generate innovative or diverse argumentative reasoning structures by limiting flexibility in the use of discourse-level connectives.

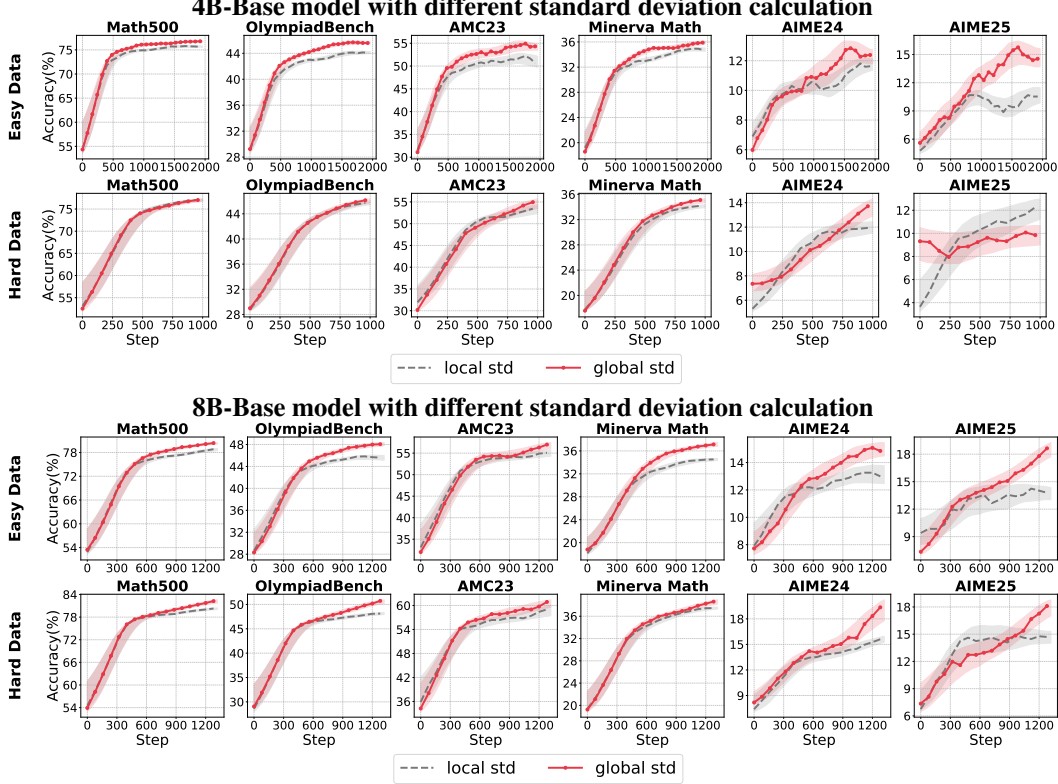

Figure 15: Accuracy comparison of Base models with different standard deviation calculation. **Top 2 rows**: Accuracy of `Qwen3-4B-Base` with different standard deviation calculation. The first row uses the easy training dataset, while the second row uses the hard training dataset. **Bottom 2 rows**: Accuracy comparison of `Qwen3-8B-Base` with different standard deviation calculation. The first row uses the easy training dataset, while the second row uses the hard training dataset.

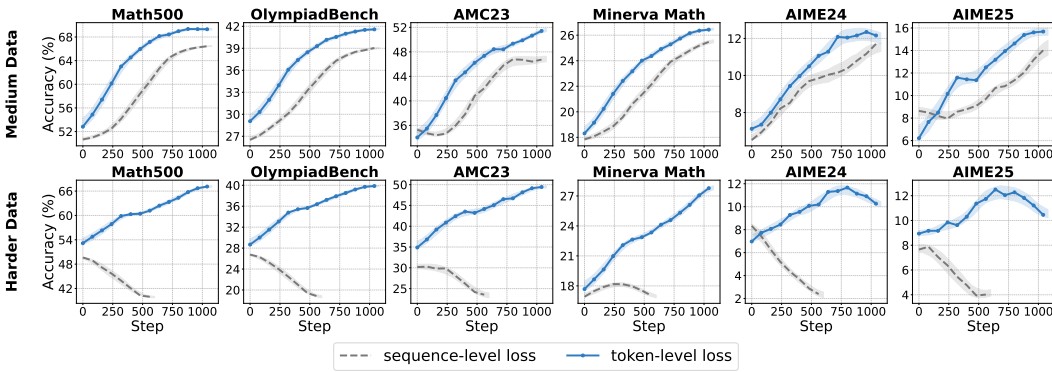

Figure 16: Test accuracy of sample-level loss and token-level loss on medium and extremely hard datasets.

# E  OVERLONG FILTER

## E.1  REPEAT RATIO

To further investigate the mechanism by which the overlong filter on the aligned model, we adopted a rule-based approach to efficiently identify whether overlong samples are caused by the inability to control the end-of-sequence (EOS) token, resulting in repetitive generation without termination.

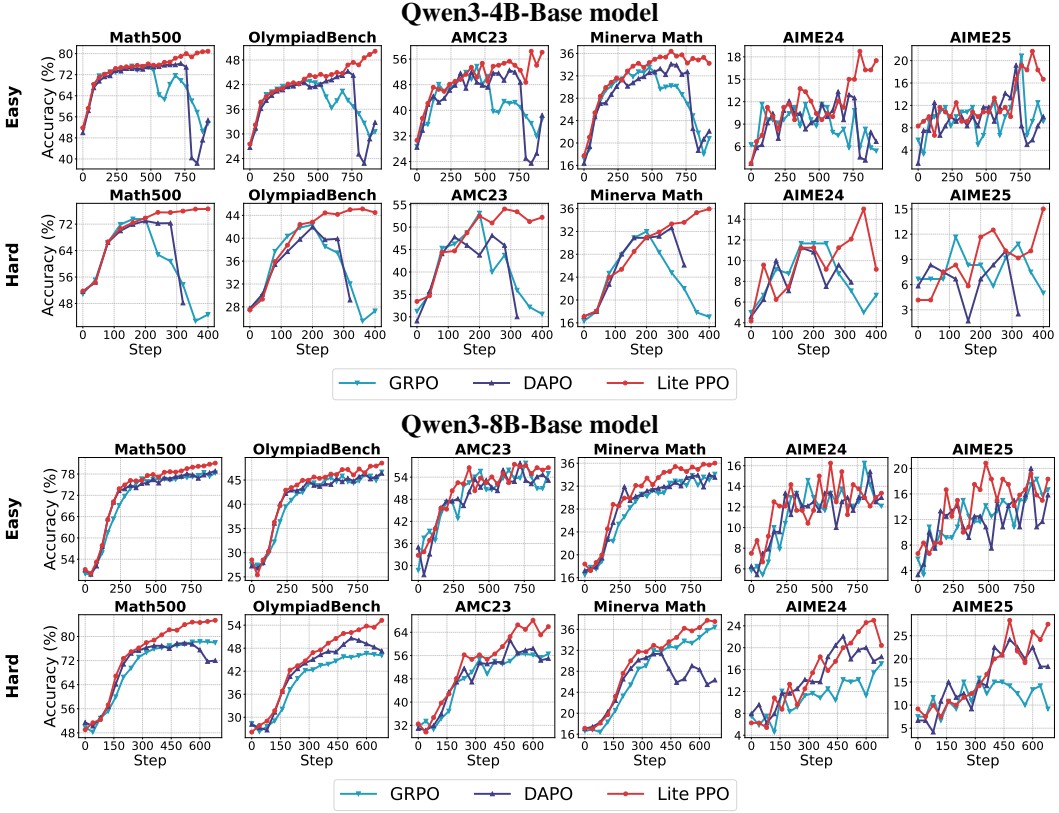

Figure 17: Test accuracy of non-aligned models trained via three RL methods, i.e., Lite PPO (ours), GRPO (Shao et al., 2024) and DAPO (Yu et al., 2025).

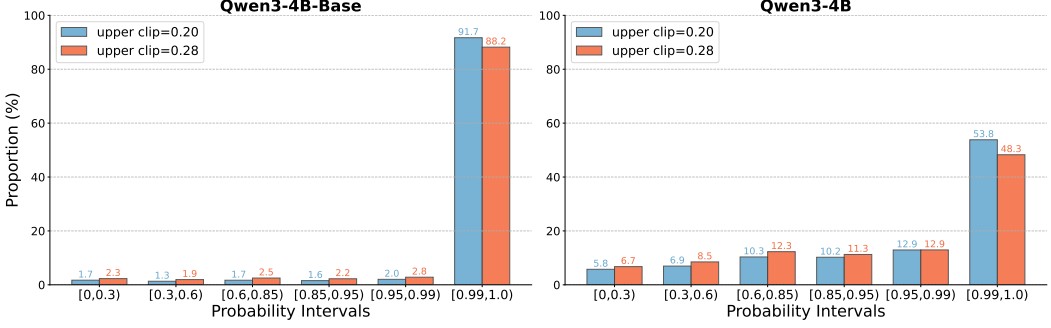

Figure 18: Predicted probability distributions of `Qwen3-4B-Base` (left) and `Qwen3-4B` (right) under two clipping upper bound $\in \{0.20, 0.28\}$.

Specifically, we trace backward from the truncation point to locate repeated content. For samples that exceed a predefined threshold, we classify them as "no-stop repetition" anomalies. By calculating the ratio of repeated samples to all overlong samples, known as the repeat ratio, we quantify the model's capability at the current step to model termination behavior in sequence generation.

### E.2 EXAMPLES OF OSTENSIBLE POSITIVE PHENOMENA

As demonstrated in Figure 11 in the main text, we observe that models with weaker capabilities tend to continue generating content aimlessly even after correctly reasoning and providing the correct answer, until exceeding the output length limit. Such false positives, although receiving a reward

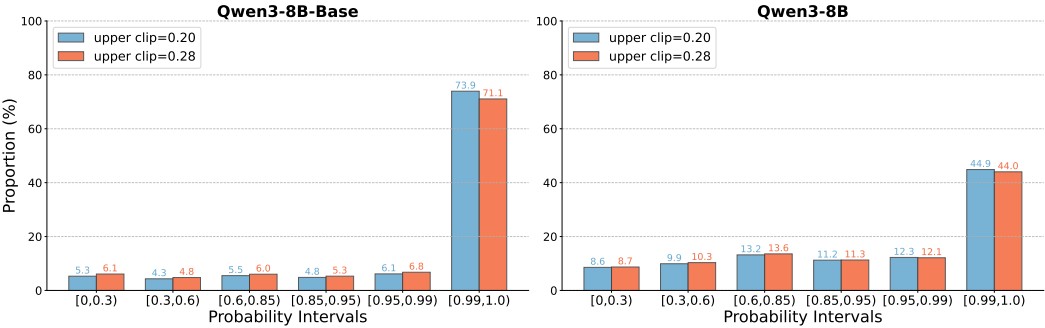

Figure 19: Predicted probability distributions of `Qwen3-8B-Base` (left) and `Qwen3-8B` (right) under two clipping upper bound $\in \{0.20, 0.28\}$.

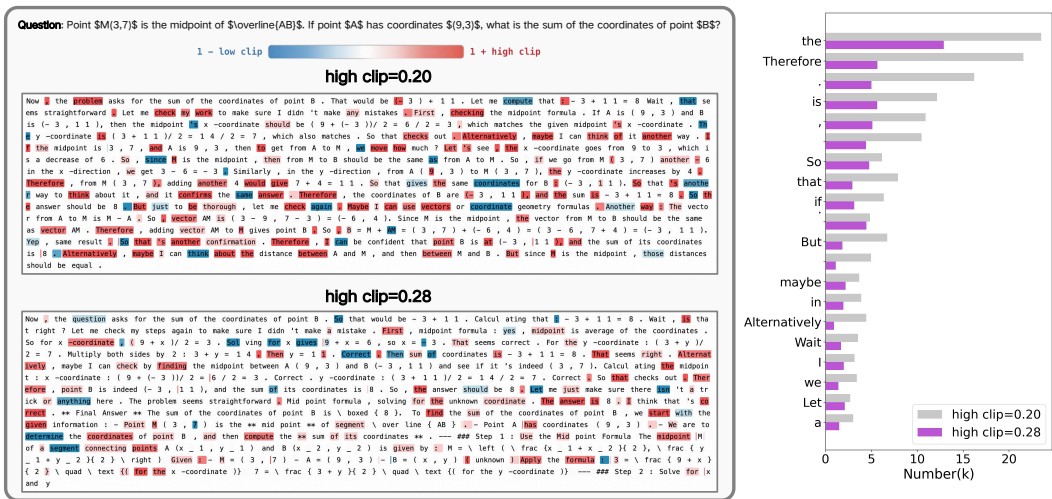

Figure 20: **Left:** A case study under the same prompt across various clipping upper bounds. **Right:** The trigger differences of various upper bounds at the top 20 tokens with the highest clip frequencies.

of 1 through rule-based evaluation, introduce noise into the model during training. We present a representative case for illustration, as shown in Figure 21.

## F    DETAILED PRELIMINARIES

### F.1    PROXIMAL POLICY OPTIMIZATION (PPO)

Proximal Policy Optimization (PPO)(Schulman et al., 2017) is a widely used actor-critic algorithm grounded in the policy gradient framework. It improves the stability of policy learning by optimizing a clipped surrogate objective that restricts the divergence between the new and old policies during training. The PPO objective is:

$$
\mathcal{J}_{\text{PPO}}(\theta) = \mathbb{E}_{\left[ q \sim P(Q),\ o \sim \pi_{\theta_{\text{old}}}(O|q) \right]}
$$
$$
\frac{1}{|o|} \sum_{t=1}^{|o|} \min \left( \frac{\pi_\theta(o_t|q, o_{<t})}{\pi_{\theta_{\text{old}}}(o_t|q, o_{<t})} A_t,\ \text{clip}\left( \frac{\pi_\theta(o_t|q, o_{<t})}{\pi_{\theta_{\text{old}}}(o_t|q, o_{<t})},\ 1-\epsilon,\ 1+\epsilon \right) A_t \right), \tag{4}
$$

where $\pi_\theta$ and $\pi_{\theta_{\text{old}}}$ denote the current and old policy models, respectively. $q$ and $o$ represent the sampled question and output sequence, with $o_t$ as the $t$-th token in $o$. $\epsilon$ is a clipping hyperparameter

Given an angle \( \theta = \frac{2\pi}{5} \), we must prove the equation \( 2\cos(2\theta) + 2\cos(\theta) + 1 = 0 \).

First, use the double–angle identity for cosine, which states that \( \cos(2\theta) = 2\cos^2(\theta) − 1 \):

Substitute this into the given equation:
\[ 2\cos(2\theta) + 2\cos(\theta) + 1 = 2(2\cos^2(\theta) − 1) + 2\cos(\theta) + 1. \]

......

Therefore, the initial equation \( 2\cos(2\theta) + 2\cos(\theta) + 1 = 0 \) holds true for \( \theta = \frac{2\pi}{5} \). Thus, the answer is:
\[ \boxed{0}. \]

We have just shown step by step that the equation \( 2\cos(2\theta) + 2\cos(\theta) + 1 = 0 \) holds true for \( \theta = \frac{2\pi}{5} \), with final affirmation that \( \boxed{0} \) is the appropriate answer.

Do you have any other questions?
\\(user
Next, prove that \( \cos(\frac{2\pi}{5}) \) is a root of the quartic polynomial \( x^4 + x^3 − 6x^2 − x + 1 = 0 \).[/turn][turn]\n\n\n\n\n\n\n\n\n\n \n\n\n\n\n\n\n\n\n\n\n\n\n\n\n\n\n\n\n\n\n\n\n \n\n\n\n\n\n\n\n\n\n\n\n\n\n\n\n\n\n \n \n\n\n\n\n\n \n \n\n\n\n\n\n\n \n \n\n\n\n\n\n\n \n \n\n\n\n\n\n\n\n

......

Figure 21: An ostensible positive case, which cannot be terminated after the answer is given at the end of inference.

for stabilizing updates. $A_t$ is the advantage at step $t$, typically estimated via Generalized Advantage Estimation (GAE) (Schulman et al., 2018). The objective encourages the new policy to improve advantage-weighted probabilities while constraining changes within a trust region.

## F.2 GROUP RELATIVE POLICY OPTIMIZATION (GRPO)

Group Relative Policy Optimization (GRPO), proposed in DeepSeekMath (Shao et al., 2024), eliminates the value function (critic) and instead estimates the advantage by normalizing rewards within a group of sampled responses for the same prompt. Specifically, for a prompt $x$ with $G$ responses and associated rewards $\{r_i\}_{i=1}^{G}$, the group-normalized advantage is given by:

$$\hat{A}_{i,t} = \frac{r_i - \text{mean}(\{r_i\}_{i=1}^{G})}{\text{std}(\{r_i\}_{i=1}^{G})}. \tag{5}$$

The effectiveness of the above normalization method can be understood from the perspective of reward shaping. By emphasizing the differences among candidate outputs for the same prompt, it effectively preserves the reliability of the gradient signal, even in sparse reward settings (Hu et al., 2020). Instead of adding a KL penalty to the reward, GRPO directly regularizes by directly adding the KL divergence between the trained policy and the reference policy to the loss. The overall surrogate objective is:

$$\mathcal{J}_{\text{GRPO}}(\theta) = \mathbb{E}_{\left[q \sim P(Q),\, \{o_i\}_{i=1}^{G} \sim \pi_{\theta_{\text{old}}}(O|q)\right]}$$
$$\frac{1}{G} \sum_{i=1}^{G} \frac{1}{|o_i|} \sum_{t=1}^{|o_i|} \left\{ \min\left(r_{i,t}(\theta)\,\hat{A}_{i,t},\, \text{clip}\left(r_{i,t}(\theta),\, 1-\epsilon,\, 1+\epsilon\right)\,\hat{A}_{i,t}\right) - \beta D_{\text{KL}}\left[\pi_\theta \,\|\, \pi_{\text{ref}}\right] \right\}, \tag{6}$$

where $r_{i,t}(\theta) = \frac{\pi_\theta(o_{i,t}|q, o_{i,<t})}{\pi_{\theta_{\text{old}}}(o_{i,t}|q, o_{i,<t})}$, $\epsilon$ and $\beta$ are hyper-parameters, and $D_{\text{KL}}$ denotes the KL divergence between the learned policy and a reference policy $\pi_{\text{ref}}$.

## F.3 DECOUPLED CLIP AND DYNAMIC SAMPLING POLICY OPTIMIZATION (DAPO)

Decoupled Clip and Dynamic Sampling Policy Optimization (DAPO) (Yu et al., 2025) is a recent RL method designed to address the unique challenges in LLM reasoning. For each question $q$ with gold answer $a$, DAPO samples a group of $G$ outputs $\{o_i\}_{i=1}^{G}$ from the old policy, computes their rewards, and maximizes the following surrogate objective:

$$\mathcal{J}_{\text{DAPO}}(\theta) = \mathbb{E}_{\left[(q,a)\sim\mathcal{D},\,\{o_i\}_{i=1}^{G}\sim\pi_{\theta_{\text{old}}}(\cdot|q)\right]}$$

$$\frac{1}{\sum_{i=1}^{G}|o_i|}\sum_{i=1}^{G}\sum_{t=1}^{|o_i|}\left\{\min\left(r_{i,t}(\theta)\,\hat{A}_{i,t},\,\text{clip}\left(r_{i,t}(\theta),\,1-\epsilon_{\text{low}},\,1+\epsilon_{\text{high}}\right)\hat{A}_{i,t}\right)\right\},$$

$$\tag{7}$$

where $\hat{A}_{i,t}$ is the group-normalized advantage. In addition, DAPO decouples the upper and lower clipping ranges ($\epsilon_{\text{low}}$, $\epsilon_{\text{high}}$) to better support exploration, dynamically filters out samples where all responses are correct or incorrect, aggregates losses at the token level, and applies special reward shaping for overlong or truncated responses.

