# OpenReview forum: "Tricks or Traps? A Deep Dive into RL for LLM Reasoning"
_ICLR.cc/2026/Conference — ICLR 2026 Poster_

### Official Review · Reviewer_MsEF · 2025-10-28

**Soundness:** 3
**Presentation:** 3
**Contribution:** 3
**Rating:** 6
**Confidence:** 3

**Summary:**

This paper shares a systematic review of variations of experimental settings in LLM-RL training to provide a unified guideline for practitioners. The review includes analysis on advantage normalization, PPO clipping and loss aggregation. In each category, the paper provides empirical results with different model sizes and dataset difficulty and a practical recommendation based on the result. Finally, the paper proposes Lite PPO, a combination of two techniques (group-mean/batch-std advantage normalization and token-level loss aggregation), improves performance of non-aligned LLM models.

**Strengths:**

The paper is well written and comprehensive. Contributions are clearly stated in introduction and supported by experiments. Addressing the lack of a standard guideline in the LLM-RL field is important to allow practitioners to understand choice of techniques. From this point of view, this paper tackles an important problem in this domain. Overall, the experimental results are well conducted and the proposed Lite PPO algorithm is an natural extension to GRPO based on the findings provided in this paper.

**Weaknesses:**

Although there isn't obvious flaws in the paper, there are comments to improve the quality of the analysis further.
- In Section 4.2.3, explanation of why 8B model doesn't have "scaling law" of the upper bound clipping parameter strengthens this analysis. Analysis on trends of LLM's outputs might help explain this.
- In Section 4.4.1, there is a lack of explanation around what leads to the experimental results shown in Figure 10 and 11. What learning dynamics influences these result? This analysis seems vital to understand the choice of overlong filtering.


Minor grammatical errors:
- Line 388, `As illustrated in As shown in` -> `As illustrated in`

**Questions:**

- Computation cost is also another important dimension when people select techniques. Do authors have any insight around this? I imagine that the most of techniques analyzed in this paper won't impact the computing performance though.

---

> ### Author Response · Authors · 2025-11-21
> **Rebuttal (Part 1)**
>
> We sincerely thank the reviewer for the insightful and positive comments, and we’d like to give a point-to-point response to address the questions and concerns.
>
> ### **Weaknesses**
>
> > **W1:** In Section 4.2.3, explanation of why 8B model doesn't have "scaling law" of the upper bound clipping parameter strengthens this analysis.
> >
>
> We appreciate this insightful observation. Our results suggest that the 4B model improves as the clipping upper bound increases, but this pattern does not hold for the 8B model. We suspect this discrpancy stems from  model scale and generation behavior.
>
> Analysis of model outputs reveals that the **8B model consistently produces longer, multi-step reasoning chains**, while the 4B model often generates truncated or single-step answers, sometimes skipping critical derivations. These differences in output behavior have direct implications for policy optimization. Because the 8B model already exhibits relatively stable and structured reasoning, its policy distribution is sharper. This means that even small parameter updates can lead to large changes in the policy ratio, especially over long sequences. When the clipping upper bound is relaxed to 0.32, these shifts are no longer sufficiently constrained, resulting in high-variance updates that can disrupt the internal logic of otherwise correct solutions.
>
> Thus, the absence of a  “scaling law”-like trend for the 8B model is not a flaw, but rather a natural consequence of its mature reasoning capability.
>
> > **W2:** In Section 4.4.1, there is a lack of explanation around what leads to the experimental results shown in Figure 10 and 11. What learning dynamics influences these result? This analysis seems vital to understand the choice of overlong filtering.
> >
>
> We have added an expanded explanation of the results presented in Figures 10 and 11. Specifically, in Figure 10, we observe performance differences under varying overlong filter lengths. After checking the response lengths, a discernible pattern emerges to explain this phenomenon. When operating under the threshold of 20k, models trained with the overlong filtering strategy exhibit a tendency to generate longer responses in comparison to the vanilla policy. Conversely, a short filter threshold, i.e., 8k, makes the model generate shorter responses.
>
> To further investigate this effect, Figure 11 (Left) shows **the distribution of clipped responses exceeding the maximum length.** Notably, in the 20k setting, both positive and negative samples are clipped more frequently due to repetitive or non-terminating outputs—a hallmark of degeneration. This indicates that, with higher length limits, the overlong mask primarily filters out unproductive or "negative" samples that contribute little to model learning. Conversely, with a stricter 8k mask threshold, the data mask filters out more samples that are long due to extended—but not necessarily degenerate—reasoning. In this setting, the model is incentivized to produce shorter, more concise responses, discouraging excessive verbosity.
>
> > **W3:** Regarding the typos and revision suggestions.
> >
>
> In the new version.We have carefully proofread the entire text and corrected this and other minor typographical issues in the revised version. Your thoughtful feedback has significantly improved the clarity and polish of our paper.

---

> > ### Author Response · Authors · 2025-11-21
> > **Rebuttal (Part 2)**
> >
> > ### **Questions**
> >
> > > **Q1:** Computation cost is also another important dimension when people select techniques. Do authors have any insight around this? I imagine that the most of techniques analyzed in this paper won't impact the computing performance though.
> > >
> >
> > Your guess is consistent with our practice!  In our implementation, all the components we study (including mixture normalization, token-level loss, and clipping variants) introduce negligible overhead compared to the base PPO pipeline. None of them require additional forward/backward passes, complex masking operations, or extra model calls. The dominant cost remains the standard policy and value network evaluations, which are shared across all variants. The results in Table 1 show the average training steps (computed over steps 150–180) across different methods and support the above conclusion.
> >
> > Moreover, we support the reviewer’s point about the need for low-cost RLHF innovations. As the field scales, we hope our work encourages more research into *efficient*, *stable*, and *minimal-overhead* RL techniques.
> >
> > Table 1. Computation time per training step across methods (average from step 150 to 180).
> >
> > | Step | actor_train/train_step | actor_train/compute_log_probs | reference/compute_log_probs |
> > | --- | --- | --- | --- |
> > | GRPO | 524.13 | 54.49 | 134.70 |
> > | DAPO | 517.59 | 56.38 | 142.14 |
> > | LitePPO | 515.98 | 54.02 | 141.47 |
> >
> > ---
> >
> > Thanks again for reviewing our work! We hope our responses address the concerns, and we would appreciate your consideration in the evaluation. We are happy to provide further clarification if needed.

---

> > > ### Author Response · Authors · 2025-11-26
> > >
> > > Dear Reviewer MsEF,
> > >
> > > Thanks once again for your time in reviewing our work and for the constructive suggestions! I would like to synchronize with you that we have re-optimized and updated the PDF according to the comments of the reviewers.
> > >
> > > As the author–reviewer discussion period concludes in less than one week, we would like to briefly follow up to confirm that our subsequent responses resolve your concerns. Any further comments would be greatly appreciated. Thanks!
> > >
> > > Best, The Authors

---

### Official Review · Reviewer_qNU9 · 2025-10-31

**Soundness:** 3
**Presentation:** 3
**Contribution:** 4
**Rating:** 6
**Confidence:** 3

**Summary:**

This paper presents a systematic and rigorous evaluation of reinforcement learning (RL) techniques for improving reasoning capabilities in large language models (LLMs). The authors address the current fragmentation in RL4LLM methodologies by reproducing and analyzing popular RL "tricks"—such as normalization, clipping, loss aggregation, and filtering—under a unified framework. Through extensive experiments across diverse model sizes (Qwen3-4B/8B), architectures (base vs. aligned), and dataset difficulties (easy/medium/hard), the study offers actionable insights into the mechanisms and applicability of each technique. A key contribution is the proposal of Lite PPO, a minimalist combination of two techniques (group-mean + batch-std normalization and token-level loss aggregation), which outperforms more complex methods like GRPO and DAPO in critic-free policy optimization.

**Strengths:**

1. Comprehensive and Reproducible Evaluation: The paper leverages a unified open-source framework (ROLL) and over 160 independent experiments, ensuring robust and statistically meaningful conclusions. The ablation studies (e.g., standard deviation removal in normalization, clip-bound scaling laws) are particularly insightful.

2. Practical Guidelines: The authors translate empirical findings into clear, scenario-specific recommendations (e.g., token-level loss for base models, sequence-level for aligned models), addressing a critical need for standardization in RL4LLM.

3. Minimalist Innovation: Lite PPO demonstrates that simplicity can outperform heavily engineered methods, challenging the trend of over-complication and offering an efficient baseline for future work.

**Weaknesses:**

1. Experiments are confined to Qwen-family models and mathematical reasoning tasks. While math is a common benchmark, broader validation on diverse domains (e.g., code generation, commonsense reasoning) would strengthen claims of generalizability.

2. While Lite PPO is promising, more ablation is needed to disentangle the contributions of its two components (normalization vs. loss aggregation) across all settings.

**Questions:**

How might your guidelines adapt to non-mathematical tasks? Are there techniques whose effectiveness is highly domain-dependent?

---

> ### Author Response · Authors · 2025-11-21
> **Rebuttal (Part 1)**
>
> We greatly appreciate the reviewer’s insightful comments and valuable feedback. We try our best to address your concerns through a large number of supplementary experiments and analysis. Please find our responses to each of the concerns below.
>
> > **W1 & Q1:** Experiments are confined to Qwen-family models and mathematical reasoning tasks. How might your guidelines adapt to non-mathematical tasks? Are there techniques whose effectiveness is highly domain-dependent?
> >
>
> We sincerely thank the reviewer for raising this important point about generalization. We have conducted additional experiments on out-of-domain reasoning benchmarks, including LCB-v5, LCB-v6, GPQA, and MMLU-Pro. As shown in Table 1, applying Lite PPO to Qwen3-8B-Base improves the average pass@1 score from **37.17**% (DAPO) to **38.45**%, confirming that our approach also transfers to non-mathematical domains.
>
> Table 1. Results on non-mathematical tasks.
>
> |  | LCB-v5 | LCB-v6 | GPQA | MMLU-Pro | Average |
> | --- | --- | --- | --- | --- | --- |
> | GRPO | 22.94 | 18.28 | 42.49 | 61.84 | 36.38 |
> | DAPO | 24.01 | 19.00 | 45.08 | 60.6 | 37.17 |
> | **LitePPO** | **25.08** | **19.71** | **46.63** | **62.38** | **38.45** |
>
> As shown in Table 2, we have conducted additional experiments on **Llama3-8B**, and following the training set up in [1], we recorded the average score among six benchmarks.
>
> Table 2. Results on llama3-8B.
>
> |  | MATH-500 | OlympiadBench | AIME 24 | AMC23 | GSM8k | Minerva Math | Average |
> | --- | --- | --- | --- | --- | --- | --- | --- |
> | GRPO | 19.8 | 8.3 | 3.7 | 18.2 | 77.3 | 8.5 | 22.63 |
> | DAPO | 25.1 | 7.6 | 4.6 | 18.4 | 77.1 | 9.2 | 23.67 |
> | Lite PPO | 27.3 | 9.7 | 9.1 | 17.9 | 79.6 | 9.6 | 25.53 |
>
> [1]:  SimpleRL-Zoo: Investigating and Taming Zero Reinforcement Learning for Open Base Models in the Wild COLM (2025)

---

> > ### Author Response · Authors · 2025-11-21
> > **Rebuttal (Part 2)**
> >
> > > **W2:** While Lite PPO is promising, more ablation is needed to disentangle the contributions of its two components (normalization vs. loss aggregation) across all settings.
> > >
> >
> > To disentangle the contributions of the two core components in Lite PPO: (1) mixture normalization and (2) token-level loss aggregation. We conducted ablation studies on **Qwen3-8B-Base** trained on the **Hard** dataset. As shown in Table 2, both components contribute meaningfully, with mixture normalization yielding a larger gain than token-level loss.
> >
> > Table 2. Ablation experiments of the LitePPO components
> >
> > |  | MATH-500 | OlympiadBench | AIME25 | MinervaMath | AIME24 | AMC23 | Average |
> > | --- | --- | --- | --- | --- | --- | --- | --- |
> > | LitePPO | **86.30** | **56.22** | **25.83** | **40.03** | **25.42** | **63.44** | **49.64** |
> > | w/o token-level loss | 80.90 | 49.29 | 14.17 | 36.31 | 16.25 | 60.00 | 42.82 |
> > | w/o mixture normalization | 69.73 | 40.82 | 15.00 | 26.61 | 15.00 | 50.00 | 36.19 |
> > | w group-level normalization | 80.05 | 47.76 | 15.83 | 36.40 | 14.58 | 56.56 | 41.86 |
> >
> > ---
> >
> > Thanks again for reviewing our work! We hope our responses address the concerns, and we would appreciate your consideration in the evaluation. We are happy to provide further clarification if needed.

---

> > > ### Author Response · Authors · 2025-11-26
> > >
> > > Dear Reviewer qNU9,
> > >
> > > Thanks once again for your time in reviewing our work and for the constructive suggestions! I would like to synchronize with you that we have re-optimized and updated the PDF according to the comments of the reviewers.
> > >
> > > As the author–reviewer discussion period concludes in less than one week, we would like to briefly follow up to confirm that our subsequent responses resolve your concerns. Any further comments would be greatly appreciated. Thanks!
> > >
> > > Best, The Authors

---

### Official Review · Reviewer_G1m5 · 2025-11-01

**Soundness:** 3
**Presentation:** 3
**Contribution:** 2
**Rating:** 4
**Confidence:** 4

**Summary:**

This paper conducts a systematic evaluation of implementation choices for RL with LLMs, covering advantage-normalization variants, clipping with a higher upper bound, loss aggregation at the token vs. sequence level, and overlong-response filtering, within a unified PPO-style framework. Experiments are run on math-reasoning benchmarks using Qwen3-4B/8B (both base and aligned variants). From these studies, the authors distill seven empirical takeaways and introduce a minimalist recipe, Lite PPO, which pairs group-mean with batch-std advantage normalization and token-level loss under a vanilla PPO objective without a critic. On base models, Lite PPO achieves consistent gains over GRPO and DAPO across several math datasets.

**Strengths:**

- Organizes scattered RL techniques into a coherent, condition-aware evaluation (base vs. aligned; easy vs. hard), yielding concrete and readable takeaways.
- Provides careful ablations and useful diagnostics (e.g., entropy and ratio behavior; token-level analyses) that improve interpretability of clipping and aggregation choices.
- Delivers a simple, reproducible recipe (Lite PPO) that reduces complexity yet performs strongly on base models, offering immediate practical value.

**Weaknesses:**

- The contribution of this paper is mainly an empirical synthesis of known implementation choices rather than an algorithmic or theoretical advance; no new RL objective or formal analysis is introduced, which constrains the paper’s originality compared to prior work
- If the practical objective is the strongest final model, the focus on base models leaves uncertain whether the proposed recipe yields meaningful gains for aligned/instruction-tuned models that start from stronger baselines.
- Evidence is confined to math reasoning; presenting the work as a general “roadmap” for RL with LLMs risks over-generalization without results on other reasoning modalities (e.g., logical, strategic, open-ended).

**Questions:**

- What motivated restricting experiments to Qwen3? Could you report at least a final-recipe run on another family (e.g., Llama or Mistral) to assess portability?
- Can you apply the Lite PPO recipe to aligned models and compare against strong aligned baselines to clarify its utility when the goal is the best final system?
- Several techniques and experimental patterns resemble DAPO. Could you delineate, in methodological terms, how your approach differs (objective, normalization, clipping, loss aggregation), and provide ablations isolating the incremental contribution beyond DAPO?
- The higher-clipping “scaling law” appears to rest on a single family with limited sweep points. Do you view this as a size-dependent trend rather than a law? If not, what additional evidence (broader sizes, denser sweeps, statistical tests) supports the stronger terminology?

---

> ### Author Response · Authors · 2025-11-21
> **Rebuttal (Part 1)**
>
> We really thank the reviewer for the insightful suggestions and in-depth useful feedback on our work. These suggestions prompted us to further refine our submission. Please find our responses to each of the concerns below.
>
> > **W1**: The contribution of this paper is mainly an empirical synthesis of known implementation choices rather than an algorithmic or theoretical advance.
> >
>
> Thanks for your questions and we will further highlight the contribution and motivation of this paper in new version.
>
> **Motivation:**
>
> The empirical analysis research on techniques has produced in-depth and extensive discussion in the classical Deep RL field [1,2,3], and they have influenced the design of subsequent reinforcement learning algorithms [4]. However, in the RL4LLM scenario, this kind of research is still in its infancy [5]. One of the key limitations is that it is difficult to draw solid practical conclusions due to the complexity of implementation, data, model, and hyperparameters. To fill this gap, this paper aims to overcome the above difficulties and provide rigorous and reliable analytical research for the RL4LLM community.
>
> **Contribution:**
>
> - **Mechanism analysis and usage guide of RLVR technology**: Aiming at the conflict between techniques and the ambiguity of mechanism ( in some cases, even seemingly similar technologies can yield **contradictory** recommendations [6].) in the design of RLVR algorithms, we for the first time from the model category. The effectiveness mechanism and applicable settings of mainstream techniques are deeply analyzed from multiple perspectives closely related to post-training practice, such as model size, dataset, generation length, and so on. Provide eight solid guiding conclusions for practitioners.
>
> - **Simple but effective algorithm：** we analyze the effectiveness of various RLVR techniques in detail and observe that there is a tendency for the algorithm design in the community to be over-technical, and advocate that the goal should be to design simple but effective user-friendly algorithms. And taking Lite PPO as a case, it is pointed out that no additional techniques need to be utilized after we effectively improve the robustness of the boost value estimation after using mixture normalization.
>
> - **Community impact**:  Critics have raised concerns about the reproducibility of RL4LLM. In this discussion, we examine various factors impacting the reproducibility of reinforcement learning (RL) algorithms, such as the sensitivity of these techniques to changes in reward scale, generation length, model series, datasets, and benchmarks. And through empirical analysis, we speculate that future research should address the question: in what contexts are these methods most effective? As a community, we should strive not only for reproducible results through fair comparisons but also to explore the best ways to demonstrate the ongoing relevance of RL.
>
> [1]: Henderson, Peter, et al. "Deep reinforcement learning that matters." *AAAI 2018*
>
> [2]: Engstrom, Logan et al. “Implementation Matters in Deep Policy Gradients: A Case Study on PPO and TRPO.” ICLR 2020
>
> [3]:  Ilyas, Andrew et al. “Are Deep Policy Gradient Algorithms Truly Policy Gradient Algorithms?” ICLR 2020
>
> [4]: Huang, Shengyi et al. “The N+ Implementation Details of RLHF with PPO: A Case Study on TL;DR Summarization.” COLM
>
> [5]: Fujimoto, Scott, Herke Hoof, and David Meger. "Addressing function approximation error in actor-critic methods." *International conference on machine learning*. PMLR, 2018.
>
> [6]: Liu, Zi-Yan et al. “Understanding R1-Zero-Like Training: A Critical Perspective.” *ArXiv* abs/2503.20783 (2025): n. pag.

---

> > ### Author Response · Authors · 2025-11-21
> > **Rebuttal (Part 2)**
> >
> > > **W2 & Q2:** Can you apply the Lite PPO recipe to **aligned models** and compare against strong aligned baselines to clarify its utility when the goal is the best final system?
> > >
> >
> > Thanks for the suggestion! We conducted additional experiments using **Qwen3-8B** as the base policy. Following our analysis in Section 4.2.1 that Clip Higher provides stronger benefits on aligned models, we instantiated a variant of LitePPO that combines mixture normalization and Clip Higher, trained on both the Easy and Hard datasets.
> >
> > As shown in Tables 1 and 2, our simple recipe significantly improves the reasoning ability of the aligned model and consistently outperforms DAPO, particularly when trained on the hard dataset. This confirms that LitePPO remains efficient for strong, instruction-tuned models.
> >
> > Table 1. Results using **Qwen3-8B (aligned model)** trained on the **Easy dataset**.
> >
> > |  | MATH-500 | OlympiadBench | MinervaMath | AIME24 | AIME25 | AMC23 | Average |
> > | --- | --- | --- | --- | --- | --- | --- | --- |
> > | DAPO | 90.57 | 64.82 | 43.43 | 47.92 | 37.54 | 79.69 | 60.66 |
> > | Lite PPO | **91.43** | **66.01** | **44.72** | **49.17** | **39.81** | **81.88** | **62.17** |
> >
> > Table 2. Results using **Qwen3-8B (aligned model)** trained on the **Hard dataset**.
> >
> > |  | MATH-500 | OlympiadBench | MinervaMath | AIME24 | AIME25 | AMC23 | Average |
> > | --- | --- | --- | --- | --- | --- | --- | --- |
> > | DAPO | 93.02 | 69.15 | 45.73 | 52.50 | 39.17 | 85.31 | 64.17 |
> > | Lite PPO | **94.77** | **71.12** | **46.60** | **49.58** | **47.50** | **92.19** | **66.96** |
> >
> > > **W3:** Evidence is confined to math reasoning; presenting the work as a general “roadmap” for RL with LLMs risks over-generalization without results on other reasoning modalities (e.g., logical, strategic, open-ended).
> > >
> >
> > Thanks for the advice! For the new version, we have conducted additional experiments to evaluate the generalization performance of our method. Specifically, we test the Qwen3-8b-Base trained by LitePPO on several out-of-domain reasoning tasks, including coding (LCB-v5 [1], LCB-v6), Interdisciplinary QA (GPQA [2]), and Language Understanding (MMLU-Pro [3]). The results in Table 3 show that Lite PPO achieves the pass@1 score of 25.08 on LCB-v5, 19.71 on LCB-v6, 46.63 on GPQA, and 62.38 on MMLU-Pro, demonstrating our method's excellent generalization ability.
> >
> > Table 3. Pass@1 score on other reasoning modalities.
> >
> > |  | LCB-v5  | LCB-v6 | GPQA | MMLU-Pro |
> > | --- | --- | --- | --- | --- |
> > | GRPO | 22.94 | 18.28 | 42.49 | 61.84 |
> > | DAPO | 24.01 | 19.00 | 45.08 | 60.6 |
> > | **LitePPO** | **25.08** | **19.71** | **46.63** | **62.38** |
> >
> > [1]: Jain, Naman et al. “LiveCodeBench: Holistic and Contamination Free Evaluation of Large Language Models for Code.” *ArXiv* abs/2403.07974 (2024): n. pag.
> >
> > [2]: Rein, David et al. “GPQA: A Graduate-Level Google-Proof Q&A Benchmark.” *ArXiv* abs/2311.12022 (2023): n. pag.
> >
> > [3]: Wang, Yubo et al. “MMLU-Pro: A More Robust and Challenging Multi-Task Language Understanding Benchmark.” *ArXiv* abs/2406.01574 (2024): n. pag.

---

> > > ### Author Response · Authors · 2025-11-21
> > > **Rebuttal (Part 3)**
> > >
> > > ### **Questions**
> > >
> > > > **Q1:** What motivated restricting experiments to Qwen3? Could you report at least a final-recipe run on another family (e.g., Llama or Mistral) to assess portability?
> > > >
> > >
> > > In the new version we have conducted additional experiments on **Llama3-8B**, and following the training set up in [1], we recorded the average score among six benchmarks, i.e., AIME 24, AMC23, GSM8k, MATH-500, Minerva Math, OlypiadBench. The results in Table 4 show that, compared to GRPO and DAPO, our Lite PPO demonstrates the best average performance, indicating the portability of our method..
> > >
> > > Table 4. Results on llama3-8B.
> > >
> > > |  | MATH-500 | OlympiadBench | AIME 24 | AMC23 | GSM8k | Minerva Math | Average |
> > > | --- | --- | --- | --- | --- | --- | --- | --- |
> > > | GRPO | 19.8 | 8.3 | 3.7 | 18.2 | 77.3 | 8.5 | 22.63 |
> > > | DAPO | 25.1 | 7.6 | 4.6 | 18.4 | 77.1 | 9.2 | 23.67 |
> > > | Lite PPO | 27.3 | 9.7 | 9.1 | 17.9 | 79.6 | 9.6 | 25.53 |
> > >
> > > [1]:  SimpleRL-Zoo: Investigating and Taming Zero Reinforcement Learning for Open Base Models in the Wild COLM (2025)
> > >
> > > > **Q3:** Several techniques and experimental patterns resemble DAPO. Could you delineate, in methodological terms, how your approach differs (objective, normalization, clipping, loss aggregation), and provide ablations isolating the **incremental contribution** **beyond** DAPO?
> > > >
> > >
> > > In the new version, we will highlight the differences and motivations of Lite PPO compared to DAPO. Methodologically, Lite PPO differs from DAPO in two key aspects:
> > >
> > > 1. **Normalization**: DAPO uses group-level normalization with both mean and variance computed within each group. However, this method of calculation has been proven to have a "length bias" that causes the model to produce long but not necessarily correct responses, reducing "token efficiency" [1].  Therefore, we propose a **mixture normalization** that combines *local (group-level) means* with *global (batch-level) variance*, which we find leads to the most stable advantage estimation.
> > > 2. **Simplicity**: Based on our mixture normalization, we find that removing several DAPO-specific components, e.g., difficulty masking, overlong response filtering, and asymmetric clipping, results in a cleaner and more portable pipeline.
> > >
> > > To isolate the incremental contribution of our design beyond DAPO, we have supplemented a set of ablation studies with the following variants (all based on Qwen3-8B-Base, trained on the easy dataset):
> > >
> > > Table 5. Ablations isolating the incremental contribution beyond DAPO. (Qwen3-8B-base, Easy dataset)
> > >
> > > |  | Normalization | Loss Aggregation | Extra tricks | MATH-500 | OlympiadBench | AIME25 | MinervaMath | AIME24 | AMC23 | Average |
> > > | --- | --- | --- | --- | --- | --- | --- | --- | --- | --- | --- |
> > > | DAPO | Group-level | Token-level | ✓ (clip, mask) | 80.20 | 49.08 | 15.00 | 35.48 | 17.50 | 60.00 | 42.87 |
> > > | DAPO w/o mask | Group-level | Token-level | ✓ (clip) | 79.35 | 47.28 | 13.33 | 35.39 | 15.83 | 58.44 | 41.60 |
> > > | DAPO w/o tricks | Group-level | Token-level | ✗ | 80.45 | 49.21 | 17.50 | 36.08 | 18.33 | 59.38 | 43.49 |
> > > | LitePPO | MixtureNorm | Token-level | ✗ | 82.42 | 50.92 | 23.33 | 39.25 | 18.75 | 61.87 | 46.09 |
> > >
> > > As shown in Table 5, LitePPO outperforms both DAPO and DAPO w/o tricks, despite using fewer components. **Notably, replacing group normalization with our mixture variant—while keeping everything else identical—leads to a consistent performance gain, confirming that the improvement stems from our normalization design rather than the removed tricks.**
> > >
> > > [1]: Liu, Zi-Yan et al. “Understanding R1-Zero-Like Training: A Critical Perspective.” *ArXiv* abs/2503.20783 (2025): n. pag.
> > >
> > > > **Q4:** The higher-clipping “scaling law” appears to rest on a single family with limited sweep points. Do you view this as a size-dependent trend rather than a law?
> > > >
> > >
> > > We treat this as a size-dependent trend and clarify it further in the new version. Our original intention was to share this phenomenon visually with readers through an interesting analogy. Strictly speaking, the "scaling law" should be formally used after more rigorous and extensive theoretical and experimental verification.
> > >
> > > ---
> > >
> > > Thanks again for reviewing our work! We hope our responses address the concerns, and we would appreciate your consideration in the evaluation. We are happy to provide further clarification if needed.

---

> > > > ### Comment · Reviewer_G1m5 · 2025-11-21
> > > >
> > > > Thank you for your detailed and thoughtful responses.
> > > > Most of my concerns have been fully addressed, and I am glad to see that the proposed approach works not only for Qwen3 but also generalizes well to other LLMs such as LLaMA.
> > > >
> > > > I am genuinely impressed by the thorough work the authors have put into the rebuttal, and I appreciate the clarity and care taken in the explanations. I intend to raise my score.
> > > >
> > > > I have just one remaining clarification:
> > > > When you refer to Qwen3-8B (aligned), do you mean the instruct version of the model? My understanding is that Qwen3-8B-base is the corresponding base model for Qwen3. Could you please confirm this?
> > > >
> > > > Finally, I would be glad to see some of the clarifications provided in the rebuttal incorporated into the final version of the manuscript (even in an appendix) so that future readers can benefit from these helpful details.

---

> > > > > ### Author Response · Authors · 2025-11-21
> > > > >
> > > > > Thank you for your kind and encouraging feedback. We sincerely appreciate your thoughtful engagement with our work.
> > > > >
> > > > > You are correct: “Qwen3-8B (aligned)” refers to the instruct version, while Qwen3-8B-Base is the corresponding base model.
> > > > >
> > > > > We will incorporate the key clarifications from the rebuttal into the final manuscript and upload an updated PDF soon.
> > > > >
> > > > > Once again, thank you for your constructive comments and support!
> > > > >
> > > > > Authors

---

### Official Review · Reviewer_rKt2 · 2025-11-10

**Soundness:** 3
**Presentation:** 3
**Contribution:** 3
**Rating:** 8
**Confidence:** 4

**Summary:**

This paper presents an empirical analysis of components of efficient RL pipelines for LLMs (in particular GRPO- and DAPO-style techniques) on both non-aligned and aligned Qwen3 models. The authors study the impact of data difficulty, advantage calculation and normalization, clipping strategies, loss aggregation granularity, and reward shaping via overlong filtering, all under a unified PPO-based setup. Based on these observations, they propose LitePPO for non-aligned models, which uses group-level mean and batch-level standard deviation for advantage normalization, together with token-level loss aggregation. Experiments on Qwen3 4B and 8B non-aligned models evaluated on Math500, OlympiadBench, AMC23, Minerva Math, AIME24, and AIME25 show that LitePPO outperforms GRPO and DAPO.

**Strengths:**

- Systematic and fairly extensive analysis of the effect of data difficulty, advantage normalization (including std/no-std variants), clipping strategies, loss aggregation, and overlong filtering on RL performance for LLMs.

- Evaluation covers both non-aligned and aligned model variants and two parameter scales (4B, 8B), which makes the conclusions more convincing than single-model studies.

- The paper is clearly written and well structured; the individual “takeaways” for each component are easy to follow and practically useful.

- The proposed LitePPO recipe is simple yet effective.

**Weaknesses:**

- Statistical robustness / variability:

    - It is not clear how many random seeds are used; Appendix C suggests a single seed (seed=42). If this is the case, the results are potentially sensitive to randomness.

    - Showing mean and variance over multiple runs (e.g., multiple seeds) would make the empirical conclusions stronger, especially in Figure 12. Even if re-running all experiments is too expensive, clearly stating the number of runs and acknowledging this limitation would help.

- Minor typo:
    - Line 388: “As illustrated in As shown in Figure 9...”, remove one of “As illustrated in / As shown in”.

**Questions:**

1. For Figures 3, 4, 6, 8, 9, 10, and 12, could you clarify precisely what the y-axis “accuracy” refers to? Is it the average accuracy across all six evaluation benchmarks, or a subset, or a particular held-out split from the Easy / Medium / Hard training datasets themselves?

---

> ### Author Response · Authors · 2025-11-21
> **Rebuttal**
>
> Thanks for your thoughtful and constructive feedback. We sincerely appreciate the time and effort you have dedicated to reviewing our work. Below, we provide a point-by-point response to your concerns:
> ### **Weaknesses**
>
> > **W1:** Concerns about the randomness of the results.
> >
>
> Thanks for raising this important point! In the new version, we ran experiments using three random seeds ({42, 52, 62}) and updated the results accordingly. Table 1 shows thatthe performance of different seeds is convergent (small standard deviation), and the overall conclusions remain consistent. All reported results are presented as **mean ± standard deviation**, using Qwen3-8B-Base trained on the Hard dataset.
>
> Table 1. Evaluation results on six mathematical reasoning benchmarks. (mean and std of Three Runs)
>
> | Method | MATH-500 | OlympiadBench | AIME25 | MinervaMath | AIME24 | AMC23 | Avg |
> | --- | --- | --- | --- | --- | --- | --- | --- |
> | GRPO | 80.01 ± 0.55 | 47.98 ± 0.67 | 15.55 ± 0.96 | 36.85 ± 0.12 | 15.13 ± 0.63 | 57.50 ± 1.07 | 42.17 ± 0.35 |
> | DAPO | 80.63 ± 1.20 | 49.37 ± 0.82 | 20.61 ± 2.49 | 35.56 ± 0.49 | 18.89 ± 1.92 | 63.61 ± 0.69 | 44.77 ± 0.33 |
> | **LitePPO** | **86.29 ± 0.47** | **55.55 ± 0.79** | **26.16 ± 2.02** | **40.51 ± 0.62** | **23.61 ± 1.27** | **64.90 ± 0.90** | **49.44 ± 0.89** |
>
> > **W2:** Minor typo: Line 388: “As illustrated in As shown in Figure 9...”, remove one of “As illustrated in / As shown in”.
> >
>
> We have carefully proofread the entire text and corrected this and other minor typographical issues.
>
> ### **Questions**
>
> > **Q1:** what the y-axis “accuracy” refers to?
> >
>
> We will highlight the meaning of the plot axis in the revised version. In Figures 3, 4, 6, 8, 9, 10, and 12, the y-axis labeled “accuracy” denotes the **arithmetic mean of the accuracies across all six evaluation benchmarks**: MATH500, OlympiadBench, MinervaMath, AIME24, AIME25, and AMC23. Specifically, for each model checkpoint or configuration, we evaluate its performance on each of these six datasets independently and then average the resulting accuracy scores to obtain the single metric plotted.
>
> ---
>
> Thanks again for reviewing our work! We hope our responses address the concerns. We are happy to provide further clarification if needed.

---

### Author Response · Authors · 2025-11-30
**Contribution summary of this research**

Thanks for your thoughtful handling of our submission and for your dedication to ensuring a fair and rigorous review process. We truly appreciate the time and effort you have invested in overseeing our paper.

We would like to briefly reiterate the core contributions of our work below.

**Contribution:** Our paper addresses this gap by conducting a systematic, unified, and open-source empirical study of key RL components—such as advantage normalization, clipping strategies, loss aggregation, and filtering—across diverse model sizes (Qwen3-4B/8B), alignment statuses (base vs. instruct), and dataset difficulties. From these analyses, we distill actionable guidelines and propose **LitePPO**, a minimalist yet highly effective recipe that consistently outperforms prior methods like GRPO and DAPO. Crucially, our work emphasizes *simplicity, reproducibility, and practical utility*—a direction we believe is essential for the sustainable advancement of RL4LLM.

### **Reviewers' recognition**
We are deeply encouraged by the reviewers’ constructive and positive feedback. All the reviewers explicitly praised different aspects of our work:

1. **Comprehensive and systematic experimental analysis** (Reviewer rKt2, Reviewer qNU9, Reviewer MsEF): We conducted systematic and extensive analysis on multiple key factors affecting LLM RL performance. Leveraging the unified open-source framework ROLL, we completed over 160 independent experiments, ensuring robust and statistically meaningful conclusions. The ablation studies (e.g., standard deviation removal in normalization, clip-bound scaling laws) are particularly insightful.
2. **Rigorous and convincing evaluation design** (Reviewer rKt2, Reviewer qNU9): Evaluation covers non-aligned/aligned models and 4B/8B scales, adopting a condition-aware framework (base/aligned; easy/hard). This makes conclusions more convincing than single-model/condition studies and organizes scattered RL techniques effectively.
3. **Strong interpretability supported by detailed diagnostics** (Reviewer G1m5): Detailed diagnostic analyses (entropy, ratio behavior, token-level) and careful ablations enhance interpretability of clipping and loss aggregation choices.
4. **Clear writing and reasonable structure** (Reviewer rKt2, Reviewer MsEF): The paper is well-written and structured, with clear contributions supported by experiments. Component-specific "takeaways" are easy to follow, facilitating grasp of key findings.
5. **Practical value with clear guidelines and reproducible recipes** (Reviewer rKt2, Reviewer G1m5, Reviewer qNU9): We provided scenario-specific recommendations (e.g., token-level loss for base models) to address RL4LLM standardization needs. The proposed LitePPO is simple, effective, reproducible, offering immediate practical value as a baseline.
6. **Meaningful minimalist innovation** (Reviewer qNU9, Reviewer MsEF): LitePPO shows simplicity can outperform complex methods, challenging over-complication. As a natural extension of GRPO, it provides a new perspective for LLM RL algorithm development.
7. **Addressing key issues in the field** (Reviewer MsEF): This work tackles the lack of standard guidelines in LLM-RL, helping practitioners choose techniques and promoting standardized development of the field.

### **Contribution during rebuttal**

In response to the concerns during the rebuttal period, we conducted extensive additional experiments, including:

1. Multi-seed evaluations (seeds {42, 52, 62}) with reported mean ± std;
2. Generalization tests on non-mathematical benchmarks (LCB-v5/v6, GPQA, MMLU-Pro);
3. Results of another family (Llama3-8B) to assess portability
4. Ablations isolating the contributions of LitePPO’s core components;
5. Experiments on aligned (instruct) models confirming LitePPO’s effectiveness beyond base models.

All these results were shared transparently in our rebuttal and have now been fully incorporated into an updated version of the manuscript.

Following our clarifications and additional experiments, Reviewer G1m5—the only initial score of 4—**raised their rating to 6 on Nov 21** (well before the Nov 27 OpenReview leakage incident) . Thus, prior to the rollback, our paper had received **fully positive and consistent evaluations (8, 6, 6, 6)** from all reviewers.

---

### Meta-Review · Area_Chair_b92w · 2026-01-07

**Summary:**

This paper studies a number of design decisions in reinforcement learning for LLM reasoning, and provides several guidelines for effective RL as well as a simplified methodology based on these guidelines that outperforms GRPO and DAPO.

**Reviewer Concerns:**

The reviewers primarily had concerns about the scope of the experiments, which focused on one model family, as well as some other missing comparisons. The reviewers provided extensive new experimental results to address these concerns.

**Reviewer Scores:**

The negative reviewer had raised their score, so all of the reviewers were positive in the end.

---

### Decision · Program_Chairs · 2026-01-26

Accept (Poster)